# Contextualize Me – The Case for Context in Reinforcement Learning

**Carolin Benjamins**[*]
*Leibniz University Hannover*
*c.benjamins@ai.uni-hannover.de*

**Theresa Eimer**[*]
*Leibniz University Hannover*
*t.eimer@ai.uni-hannover.de*

**Frederik Schubert**
*Leibniz University Hannover*
*schubert@tnt.uni-hannover.de*

**Aditya Mohan**
*Leibniz University Hannover*
*a.mohan@ai.uni-hannover.de*

**Sebastian Döhler**
*Leibniz University Hannover*
*doehler@tnt.uni-hannover.de*

**André Biedenkapp**
*University of Freiburg*
*biedenka@cs.uni-freiburg.de*

**Bodo Rosenhahn**
*Leibniz University Hannover*
*rosenhahn@tnt.uni-hannover.de*

**Frank Hutter**
*University of Freiburg*
*fh@cs.uni-freiburg.de*

**Marius Lindauer**
*Leibniz University Hannover*
*m.lindauer@ai.uni-hannover.de*

**Reviewed on OpenReview:** *https://openreview.net/forum?id=Y42xVBQusn*

## Abstract

While Reinforcement Learning (RL) has made great strides towards solving increasingly complicated problems, many algorithms are still brittle to even slight environmental changes. Contextual Reinforcement Learning (cRL) provides a framework to model such changes in a principled manner, thereby enabling flexible, precise and interpretable task specification and generation. Our goal is to show how the framework of cRL contributes to improving zero-shot generalization in RL through meaningful benchmarks and structured reasoning about generalization tasks. We confirm the insight that optimal behavior in cRL requires context information, as in other related areas of partial observability. To empirically validate this in the cRL framework, we provide various context-extended versions of common RL environments. They are part of the first benchmark library, CARL, designed for generalization based on cRL extensions of popular benchmarks, which we propose as a testbed to further study general agents. We show that in the contextual setting, even simple RL environments become challenging - and that naive solutions are not enough to generalize across complex context spaces.

---

[*] Equal Contribution

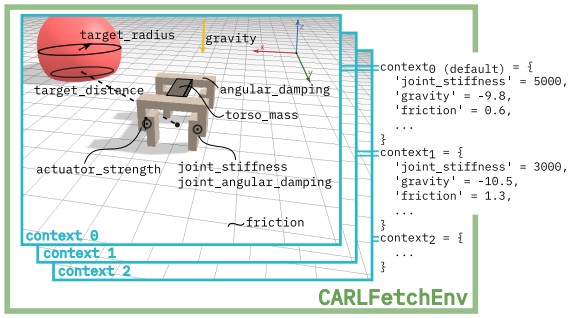

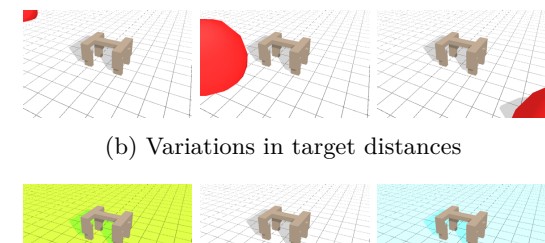

(b) Variations in target distances

(a) Example of a contextual extension of Brax' Fetch (Freeman et al., 2021) as part of `CARL`

(c) Ground friction simulating grass, concrete and ice

Figure 1: `CARL` allows to configure and modify existing environments through the use of context. This context can be made visible to agents through context features to inform them directly about the current instantiation of the context (see a). Specific instances of CARLFetch with variations in goal distance (see b) and with different ground frictions (grass, concrete and ice, see c).

# 1 Introduction

Reinforcement Learning (RL) has shown successes in a variety of domains, including (video-)game playing (Silver et al., 2016; Badia et al., 2020), robot manipulation (Lee et al., 2020a; Ploeger et al., 2020), traffic control (Arel et al., 2010), chemistry (Zhou et al., 2017), logistics (Li et al., 2019) and nuclear fusion (Degrave et al., 2022). At the same time, RL has shown little success in real-world deployments that require generalization, rather focusing on narrow domains (Bellemare et al., 2020; Degrave et al., 2022). We believe this can largely be explained by the fact that modern RL algorithms are not designed with generalization in mind, making them brittle when faced with even slight variations of their environment (Yu et al., 2019; Meng & Khushi, 2019; Lu et al., 2020).

To address this limitation, recent research has increasingly focused on generalization capabilities of RL agents. Ideally, general agents should be capable of zero-shot transfer to previously unseen environments and robust to changes in the problem setting while interacting with an environment (Ponsen et al., 2009; Henderson et al., 2018; Cobbe et al., 2020; Zhang et al., 2021b; Fu et al., 2021b; Yarats et al., 2021; Abdolshah et al., 2021; Sodhani et al., 2021b; Adriaensen et al., 2022; Kirk et al., 2023). Steps in this direction have been taken by proposing new problem settings where agents can test their transfer performance, e.g. the Arcade Learning Environment's flavors (Machado et al., 2018) or benchmarks utilizing Procedural Content Generation (PCG) to increase task variation, e.g. ProcGen (Cobbe et al., 2020), NetHack (Küttler et al., 2020) or Alchemy (Wang et al., 2021). Furthermore, robustness to distribution shift as well as multi-task learning have been long-standing topics in meta-RL, both in terms of benchmarks (Yu et al., 2019; Sodhani et al., 2021a) and solution methods (Pinto et al., 2017; Finn et al., 2017; Zhu et al., 2020; Zhang et al., 2021d).

While these extended problem settings in RL have expanded the possibilities for benchmarking agents in diverse environments, the degree of task variation is often either unknown or cannot be controlled precisely. We believe that generalization in RL is held back by these factors, stemming in part from a lack of problem formalization (Kirk et al., 2023). In order to facilitate generalization in RL, cRL proposes to explicitly take environment characteristics, the so called *context* (Hallak et al., 2015), into account. This inclusion enables precise design of train and test distributions with respect to this context. Thus, cRL allows us to reason about which types of generalization abilities RL agents exhibit and to quantify their performance on them. Overall, cRL provides a framework for both theoretical analysis and practical improvements.

In order to empirically study cRL, we introduce a benchmark library for Context-Adaptive Reinforcement Learning: `CARL`. `CARL` collects well-established environments from the RL community and extends them with the notion of context. To ensure interpretability, `CARL` considers context which is mainly based on physical properties and thus intuitive to humans. For example, `CARL` extends Brax (Freeman et al., 2021) environments with properties such as friction, gravity, or the mass of an object (see Figure 1). Through `CARL`'s interface, it is possible to meticulously define the context distributions on which RL agents are trained and evaluated. We

use our benchmark library to empirically show how different context variations can significantly increase the difficulty of training RL agents, even in simple environments. We further verify the intuition that allowing RL agents access to context information is beneficial for generalization tasks in theory and practice.

In short, our contributions are: (i) We provide a theoretical and empirical overview of why Contextual Reinforcement Learning (cRL) is useful for research into zero-shot generalization for RL; (ii) We introduce our benchmark library `CARL` which enables fine-grained context control in benchmarking cRL; and (iii) We demonstrate that even on simple environments, generalization is challenging for standard RL agents.

## 2  Contextual Markov Decision Processes

In order to facilitate generalization, we first have to rethink how we model the RL problem. While we could follow the common notion of modeling environments as Markov Decision Processes (MDPs), this way of modeling typically assumes a single, clearly defined environment. We believe this problem formulation is overly restrictive. Agents trained under such an assumption fail when the underlying environment does not behave exactly as they have experienced during training. Modeling the problem as a contextual MDP (cMDP) following Hallak et al. (2015); Modi et al. (2018) instead, we assume that there are multiple related but distinct environments which an agent might interact with and which can be characterized through *context*. This notion of context provides us with the means necessary to study the generalization abilities of RL agents in a principled manner.

**What is Context?** To help build intuition on what context is and how it might influence the learning problem, we first give an informal treatment of context. In essence, context characterizes how the environment behaves and what its goals look like. In contrast to the observations of an MDP, which describe the changes to the environment step by step, context allows us to reason about how the state will evolve without requiring access to the true transition and reward functions. Further, context features are typically static (i.e., do not change during an episode) or change at a much slower time scale than state observations.

In a robot, for example, joint friction could inform an RL controller how much torque to apply to execute some desired action. In the short horizon, the friction will not change. However, especially if the robot is not well maintained, the friction can increase due to mechanical degradation with the need to adapt accordingly. Context information can now help to appropriately compensate. Another example could be different payloads as context for a robot or different winds for a helicopter (Koppejan & Whiteson, 2009). Note that such a feature does not need to provide the exact change in transition dynamics, rather it needs to provide a signal of how transition (and reward) functions relate to each other.

Context does not need to influence reward and transition functions at the same time. In goal-based reinforcement learning (e.g., Schaul et al., 2015; Eysenbach et al., 2019), the notion of goals influences the reward function, typically without changing the transition dynamics. For example, an agent needs to traverse an empty gridworld. A goal that is placed to the right of the agent yields high rewards by moving closer. If another goal is now on the left of the agent, the reward for actions switches without changing how the agent traverses the grid (i.e. the transition function).

By its nature, context enables agents to learn more discriminative and thus more general policies. This makes context very well suited to study generalization of RL agents (Kirk et al., 2023). However, context has yet to be widely explored or leveraged in reinforcement learning, and there are many open questions to be addressed. A prominent one among them is how to use context during learning. We discuss contextual RL more formally in Section 4.

**Contextual Markov Decision Processes (cMDP)** (Hallak et al., 2015; Modi et al., 2018) allow us to formalize generalization across tasks by extending the standard definition of an MDP in RL. An MDP $M = (\mathcal{S}, \mathcal{A}, \mathcal{T}, \mathcal{R}, \rho)$ consists of a state space $\mathcal{S}$, an action space $\mathcal{A}$, transition dynamics $\mathcal{T}$, a reward function $\mathcal{R}$ and a distribution over the initial states $\rho$. Through the addition of context, we can *define, characterize* and *parameterize* the environment's rules of behavior and therefore induce task *instances* as variations on the problem. In the resulting cMDP, the action space $\mathcal{A}$ and state space $\mathcal{S}$ stay the same; only the transition dynamics $\mathcal{T}_c$, the reward $\mathcal{R}_c$ and the initial state distribution $\rho_c$ change depending on the context $c \in \mathcal{C}$. Through the context-dependent initial state distribution $\rho_c$, as well as the change in dynamics, the agent may

furthermore be exposed to different parts of the state space for different contexts. The context space $\mathcal{C}$ can either be a discrete set of contexts or defined via a context distribution $p_{\mathcal{C}}$. A cMDP $\mathcal{M}$ therefore defines a set of contextual MDPs $\mathcal{M} = \{M_c\}_{c \sim p_{\mathcal{C}}}$.

**cMDPs Subsuming Other Notions of Generalization** Even though a lot of work in RL makes no explicit assumptions about task variations and thus generalization, there are extensions of the basic MDP models beyond cMDPs that focus on generalization. One of these is *Hidden-Parameter MDPs* (Doshi-Velez & Konidaris, 2016), which allows for changes in dynamics just as in cRL, but keeps the reward function fixed. The reverse is true in goal-based RL (Florensa et al., 2018), where the reward function changes, but the environment dynamics stay the same. *Block MDPs* (Du et al., 2019) are concerned with a different form of generalization than cMDPs altogether; instead of zero-shot policy transfer, they aim at learning representations from large, unstructured observation spaces. An alternative approach is the *epistemic POMDP* (Ghosh et al., 2021) as a special case of a cMDP. Here, transition and reward functions may vary, but the context is assumed to be unobservable. The corresponding approaches then model the uncertainty about which instance of the cMDP the agent is deployed on during test time.

Settings that can be described as a collection of interacting systems, e.g. multi-agent problems, can be formalized as such through *Factored MDPs* (Boutilier et al., 1999; Guestrin et al., 2001). The generalization target here is again not necessarily zero-shot policy transfer, but generalization with respect to one or more of the factored components, e.g. the behavior of a competing agent. Both Block MDPs and Factored MDPs are compatible with cMDPs, i.e., we can construct a Block cMDP or a Factored cMDP in order to focus on multiple dimensions of generalization (Sodhani et al., 2021a).

Apart from these MDP variations, there are also less formalized concepts in RL related to or subsumed by cMDPs. Skill-based learning, for example, relates to cMDPs (da Silva et al., 2012). Some variations of the environment, and therefore areas of the context space, will require different action sequences than others. The past experience of an agent, i.e. its memory, can also be seen as a context, even though it is rarely treated as such. Frame stacking, as is common in, e.g., Atari (Bellemare et al., 2016) accomplishes the same thing, implicitly providing context by encoding the environment dynamics through the stacked frames.

**Obtaining Context Features** Not every task has an easily defined or measurable context that describes the task in detail. Therefore, it is important to examine how context can be obtained in such cases. Often we can only extract very simple context features for a task. For example, even though procedural generation (PCG) based environments do not allow control over the training and test distributions, they can still be considered contextual environments since they are usually seeded. This would give us the random seed as context information (Kirk et al., 2023). However, the seed provides no semantic information about the instance it induces. Obtaining more useful context features should therefore be a focus of cRL. Learned representations provide an opportunity (Jaderberg et al., 2017b; Gelada et al., 2019; Zhang et al., 2021a; Castro et al., 2021) to do this in a data-driven manner. As these representations encode the tasks the agent needs to solve, subspaces of the context space requiring different policies should naturally be represented differently. This idea has previously been applied to detecting context changes in continuous environments (da Silva et al., 2006; Alegre et al., 2021) and finding similar contexts within a training distribution (da Silva et al., 2012). Thus, even without readily available context, representation learning can enable the reliable availability of information relevant for generalization to tasks both in- and out-of-distribution.

## 3 Related Work

Transferring and generalizing the performance of an RL agent from its training setting to some test variation has been at the center of several sub-communities within RL. Robustness, for example, can be seen as a subcategory of generalization where variations to the context are usually kept small, and the goal is to avoid failures due to exceptions on a single task (Morimoto & Doya, 2000; Pinto et al., 2017; Mehta et al., 2019; Zhang et al., 2021d). Policy transfer is also concerned with generalization in a sense, though here the goal has often been fine-tuning a pre-trained policy, i.e., few-shot generalization instead of zero-shot generalization (Duan et al., 2016; Finn et al., 2017; Nichol et al., 2018). The goal in Multi-Task Learning is to learn a fixed set of tasks efficiently (Yu et al., 2019), not necessarily being concerned with generalizing outside of this set. Meta-Learning in RL usually aims at zero-shot policy generalization similar to Contextual

Reinforcement Learning (cRL). This field is very broad with different approaches like learning to learn algorithms or their components (Schulman et al., 2016; Duan et al., 2016; Wang et al., 2017), generating task curricula (Matiisen et al., 2020; Nguyen et al., 2021) or meta-learning hyperparameters (Runge et al., 2019; Zhang et al., 2021c).

The previously mentioned methods were not conceived with cRL in mind but use context implicitly. Many meta-RL methods, however, can or do make use of context information to guide their optimization, either directly (Klink et al., 2020; Eimer et al., 2021) or by utilizing a learnt dynamics model (Kober et al., 2012). The idea of context-aware dynamics models has also been applied to model-based RL (Lee et al., 2020b). These approaches use context in different ways to accomplish some generalization goal, e.g. zero-shot generalization to a test distribution or solving a single hard instance. In contrast, we do not propose a specific Meta-Learning method but examine the foundations of cRL and how context affects policy learning in general.

Zero-shot generalization across a distribution of contexts, specifically, has become a common goal in standard RL environments, often in the form of generalization across more or less randomly generated tasks (Juliani et al., 2019; Cobbe et al., 2020; Samvelyan et al., 2021). The larger the degree of randomness in the generation procedure, however, the less context information and control are available for Meta-Learning methods (Kirk et al., 2023). Such underspecification of tasks can even make evaluations more challenging (Jayawardana et al., 2022). In contrast to other works on generalization in RL, we therefore focus on sampling context without PCG but from explicitly defined distributions. This allows us to analyze the capabilities of our agents in a more fine-grained manner, e.g. how far away from their training distribution generalization performance starts to decrease, instead of relying only on the test reward across all task instances. Our benchmark is styled similarly to the early generalized helicopter environment, where Koppejan & Whiteson (2009) can control and vary wind. Later, Whiteson et al. (2011) propose an evaluation protocol for general RL to avoid overfitting on particular training environments, where they argue for generalized methodologies assessing the performance of an agent on a set or distribution of environments. cMDPs easily fit into this line of evaluation protocols with the advantage of interpretable generalization capabilities because of the definition of interpretable context features. Kirk et al. (2021) similarly propose evaluation protocols, but already with cMDPs in mind. For further ways cRL opens new directions in ongoing work, see Appendix G.

# 4  Reinforcement Learning with Context

In this section, we provide an overview of how context can influence the training of RL agents. We discuss training objectives in the contextual setting and give a brief theoretical intuition of the implications of treating generalization problems that can be modeled as cMDPs like standard MDPs. This should serve as a demonstration of why using the cMDP framework for generalization problems is beneficial and should be explored further. Note that we assume standard cMDPs in this section, meaning the context, if provided to the agent, is fully observable and reflects the true environment behavior.

## 4.1  Solving cMPDs

**Objectives** Having defined context and cMDPs, we can now attempt to solve cMDPs. To this end, we must first formulate potential **objectives** being addressed by cRL. In contrast to standard RL, cRL offers several different objectives for the same training setting depending on what kind of generalization we are aiming for. We can provide this objective via a target context distribution that induces the target cMDP $\mathcal{M}$. Depending on the relation between target and training distributions, we can measure interpolation performance, robustness to distribution shift, out-of-distribution generalization, and more, e.g., solving a single expensive hard task by only training on easy ones. Thus, the cRL objective is *defined by the relationship between train and test settings*, similar to supervised learning, but on the level of tasks rather than the level of data points (see Figure 2).

**Optimality** Regardless of the specific objective, we solve cMDPs in the same way we would solve standard MDPs, though we need to extend the definition of the return. Instead of maximizing the expected reward over time, we use the expected reward over both time and target context distribution. Therefore we define **optimality** in a cMDP in the following way:

Figure 2: Different train and test relationships result in different generalization tasks: interpolation between known friction levels (left), generalizing to goals further away than seen in training (middle), generalizing to the goal distances in the training distribution with lower friction (right).

**Definition 1** *A policy $\pi^*$ is optimal for a given cMDP $\mathcal{M}$ with target context distribution $p_{\mathcal{C}}$ iff $\pi^*$ optimally acts on every MDP in $\mathcal{M}$ (i.e. maximizes the return $G_{c,\pi}$ for each context c):*

$$\forall c \sim p_{\mathcal{C}} : \pi^* \in \arg\max_{\pi \in \Pi} \mathbb{E}_\pi[G_{c,\pi}]$$

Note that such an optimal policy does not necessarily exist. Malik et al. (2021) showed that the problem of learning a policy $\pi$ that satisfies Definition 1 may be intractable for some context distributions, even if the contexts are similar.

In order to compare policies across a given target distribution, we propose to compare the gap between optimal and actual performance, what we call the *Optimality Gap $\mathcal{OG}$*. Formally, we define $\mathcal{OG}$ as the gap between the optimal return $G_{c,\pi^*}$ over the target context distribution $c$ and the return of the given policy $\pi$:

$$\mathcal{OG} := \mathbb{E}_{p_{\mathcal{C}}}[G_{c,\pi^*} - G_{c,\pi}]. \tag{1}$$

In settings where the optimal return is known, we can directly evaluate the optimality gap as the difference between the return of a trained policy and the optimal return. However, in cases where the optimal return is either unknown or intractable, we can instead use an agent trained on each single context as an approximation of the optimal return. However, we have two sources of uncertainty: First, the specialized agent might not reach the best performance achievable in the MDP and second, the uncertainty of the optimal return also depends on the number of context samples for the specialized agent.

## 4.2 Optimal Policies Require Context

In this section, we give an intuition and a proof sketch of why conditioning the policy not only on the state space $\mathcal{S}$ but also on the context space $\mathcal{C}$ can be beneficial. This is very much analogous to how a larger degree of observability will enable better learning in POMDPs (Kurniawati, 2022). As a reminder, a standard RL policy is defined as a mapping $\pi : \mathcal{S} \to \mathcal{A}$ from a state observation $s \in \mathcal{S}$ to an action $a \in \mathcal{A}$.[1] In contrast to standard RL, in cRL the state space $\mathcal{S}$ is shared among all MDPs within a cMDP, meaning that states can occur in multiple MDPs even though the transition and reward functions might differ.

**3-State cMDP** A simple way to exemplify this is through the 3-state cMDP in Figure 3. For the first MDP on the left, the optimal action would be $a_0$ leading to state $S_1$ with a high reward of 10. The MDP in the middle is a variation of the same state and action space, where the transition function has changed: while the reward in state $S_1$ remains 10, action $a_0$ now leads to state $S_2$ with a lower reward of 1. Similarly, the MDP on the right is another variation changing the reward function instead of the transition function: $a_0$ still leads to $S_1$, but the associated reward now is 1. An agent exposed to such changes would not be able to react appropriately unless the policy is conditioned on the context $c_i$.

In contrast, a *context-conditioned policy* can distinguish between the contexts and thus receives more guiding feedback during training. Also, it can be more capable to perform optimally at test time given an approximation of the context. We define context-conditioned policies as follows:

---

[1]Alternatively, one can also define a policy as a probability distribution P(a|s) over the actions given a state. The following line of arguments also holds for stochastic policies but we only outline it for deterministic policies to not clutter notation.

**Definition 2** *A context-conditioned policy is a mapping $\pi : \mathcal{S} \times \mathcal{C} \to \mathcal{A}$ with state space $\mathcal{S}$, context space $\mathcal{C}$ and action space $\mathcal{A}$.*

In order to formalize this intuitive explanation of why context is helpful in generalization, let us first recall what optimal performance means in cMDPs. Definition 1 requires that there exists a policy $\pi^*$ that is optimal on every context. We will now show that optimal context-oblivious policies like this do not exist in general, but that to obtain optimality, the policy needs to have access to the context.

**Proposition 1** *For a given cMDP $\mathcal{M} = \{M_{c_1}, M_{c_2}\}$ defined over two possible contexts $c_1$ and $c_2$, there is either a common (context-oblivious) optimal policy $\pi^*$ for both contexts or there is at least one conflict state $s'$ at which the optimal policy differs between the contexts.*

**Proof Sketch** Let us look at a given state $s \in \mathcal{S}$ that is reachable in $M_{c_1}$. Further, let us assume $\pi^*_{c_1}$ is the optimal policy for $M_{c_1}$ that is defined only on states reachable in $M_{c_1}$. We have to consider the following three possible cases for each state $s \in \mathcal{S}$: (i) $s$ is reachable in $M_{c_2}$ and $\pi^*_{c_1}(s)$ is not optimal on $M_{c_2}$; (ii) $s$ is reachable in $M_{c_2}$ and $\pi^*_{c_1}(s)$ is optimal on $M_{c_2}$; (iii) $s$ is not reachable in $M_{c_2}$. We assume that we act within one MDP.

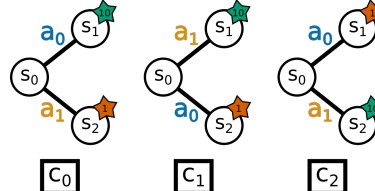

If (i) is true for at least one state $s \in \mathcal{S}$, the optimal policy obviously is different between the contexts in this state, and therefore we have found a conflict state $s'$. The other two cases do not produce such a conflict. We can, however, construct a policy $\pi^*$ that is optimal on $M_{c_1}$ and $M_{c_2}$ from $\pi^*_{c_1}$ if for all $s \in \mathcal{S}$ either (ii) or (iii) is true. For any state $s$ where (ii) holds, we simply set $\pi^*(s) = \pi^*_{c_1}(s)$. For states $s$ that are not reachable

Figure 3: A sample cMDP with three contexts. The original one (left), one changing the transition function (middle) and another the reward function (right).

in $M_{c_1}$ as stated in (iii), $\pi^*_{c_1}$ is not defined. We can therefore extend $\pi^*$ by these states without changing its optimality on $M_{c_1}$. Let $a^*$ be the optimal action in such $s$ on $M_{c_2}$. Then, we define $\pi^*(s) = a^*$. By construction, $\pi^*$ is then optimal on all states $s \in \mathcal{S}$, and it exists iff there is no state reachable in both contexts where the optimal action for $M_{c_1}$ differs from the one for $M_{c_2}$. ∎

**Theorem 1** *An optimal policy $\pi^*$ for any given cMDP $\mathcal{M}$ is only guaranteed to exist if it is conditioned on the context: $\pi : \mathcal{S} \times \mathcal{C} \to \mathcal{A}$.*

**Proof Sketch** Let us assume we know an optimal policy $\pi^*_c$ for any context $c \in \mathcal{C}$. As $c$ induces an MDP and with it a corresponding optimal policy, we know that $\pi^*_c$ exists. Furthermore, let the sets of optimal policies between at least two MDPs $M_{c_1}$ and $M_{c_2}$ be disjoint. This means no policy exists that is optimal on both $c_1$ and $c_2$.

Now let us examine the optimal policy $\pi^*$ and assume that it exists for this cMDP $\mathcal{M}$. By definition, $\pi^*$ is optimal on $c_1$ and $c_2$. If it is *only* conditioned on the state, $\pi^*(s)$ results in the same action independent of the context. Because the sets of optimal policies for $c_1$ and $c_2$ are disjoint, there must be at least one state $s'$, where $\pi^*_{c_1}(s') \neq \pi^*_{c_2}(s')$ according to Proposition 1. As both are optimal in their respective contexts but not in the other and do not result in the same action for $s$, $\pi^*$ cannot actually be optimal for both $c_1$ and $c_2$. Thus, the optimal policy $\pi^*$ does not exist. On the other hand, if we can condition the policy on the context, such that $\pi : \mathcal{S} \times \mathcal{C} \to \mathcal{A}$, we can circumvent this problem (for a discussion on how this relates to partial observability, see Appendix A). $\pi^*(s, c) = \pi^*_c$ is optimal for each context. ∎

**Discussion** Theorem 1 raises the question of how performance changes if the policy is *not* conditioned on the context. Apart from the fact that we can construct cMDPs where a policy not conditioned on the context may perform arbitrarily poorly, we intuitively assume the optimality gap should grow the broader $p_{\mathcal{C}}$ becomes and the more impact slight changes in $c$ have on the transitions and rewards. Formally assessing the optimality gap $\mathcal{OG}$ is another challenge in itself. Another question is how relevant the assumption of disjoint sets of optimal policies for different contexts is in practice. For the case where the observations implicitly encode the context, a single policy can solve different contexts optimally by never entering the conflict above. Assuming this is how generalization is commonly handled in RL. However, we deem this

not to be a reliable mechanism to avoid conflicts between context-optimal policies on the same observations. In conclusion, depending on the environment and on how context reflects on the observations, it might be beneficial to explicitly include context, especially on harder and more abstract generalization tasks.

## 5 The `CARL` Benchmark Library

To analyze how the context and its augmentation influence the agent's generalization capabilities, learning, and behavior, we propose `CARL`: a library for *Context Adaptive Reinforcement Learning* benchmarks following the Contextual Reinforcement Learning formalism. In our release of `CARL` benchmarks, we include and contextually extend classic control and box2d environments from OpenAI Gym (Brockman et al., 2016), Google Brax' walkers (Freeman et al., 2021), a selection from the DeepMind Control Suite (Tassa et al., 2018), an RNA folding environment (Runge et al., 2019) as well as Super Mario levels (Awiszus et al., 2020; Schubert et al., 2021), see Figure 4.

**Benchmark Categories** Often the physics simulations (brax, box2d, classic control and dm control) define a dynamic body in a static world with similar physical entities like gravity, geometry of the moving body and mass. In our example CARLFetch from Figure 1a the goal is to move the agent to the target area. The context features joint stiffness, gravity, friction, (joint) angular damping, actuator strength, torso mass as well as target radius and distance define the context and influence the exact instantiation and dynamics of the environment. In principle, the designer is free to select the context features from the set of parameters defining the environment. For Fetch, this could also be limb length of the body. Out of practicality, we choose to vary the most common physical attributes across the environments. When selecting an environment's parameter to become a context feature, it must be guaranteed that the physics and the environment's purpose are not violated, e.g. by setting a negative gravity such that the body flies up and is never able to reach the goal. Please see Appendix H for all registered context features per environment.

Besides physical simulation environments, `CARL` provides two more specific, challenging environments. The first is the CARLMarioEnv environment built on top of the TOAD-GAN level generator (Awiszus et al., 2020; Schubert et al., 2021). It provides a procedurally generated game-playing environment that allows customization of the generation process. This environment is therefore especially interesting for exploring representation learning for the purpose of learning to better generalize. Secondly, we move closer to real-world application by including the CARLRNADesignEnvironment (Runge et al., 2019). The challenge here is to design RNA sequences given structural constraints. As two different datasets of

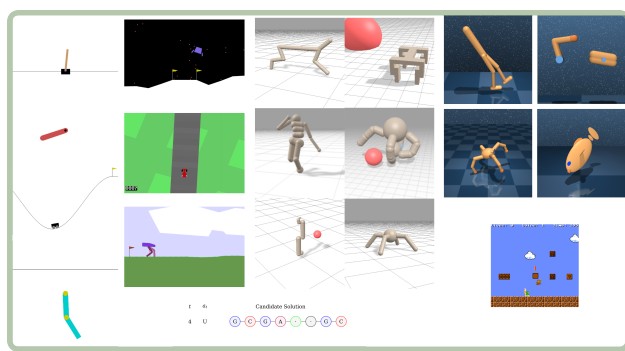

Figure 4: The `CARL` benchmarks

structures and their instances are used in this benchmark, it is ideally suited for testing policy transfer between RNA structures.

### 5.1 Properties of Benchmarks

While the categorization of the `CARL` benchmarks above provides an overview of the kinds of environments included, we also discuss them in terms of relevant environment attributes that describe the nature of their problem setting, see Figure 5.

**State Space** Most of our benchmarks have vector-based state spaces, allowing to concatenate context information. Their sizes range from only two state variables in the CARLMountainCar environments to 299 for the CARLHumanoid environment. The notable exceptions here are CARLVehicleRacing and CARLToadGAN, which exclusively use pixel-based observations.

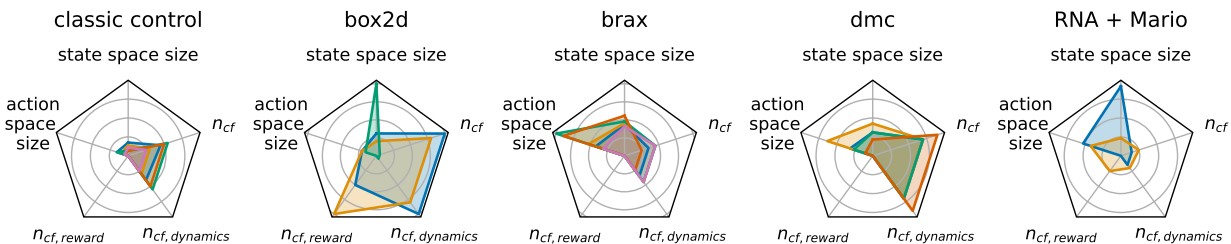

Figure 5: Characteristics of each environment of the environment families showing the action space size, state space size (log-scale), number of context features ($n_{cf}$), the number of context features directly shaping the reward ($n_{cf,reward}$) and the ones changing the dynamics ($n_{cf,dynamics}$). All axes are scaled to the global extrema and the state space size is additionally on a logarithmic scale.

**Action Space** We provide both discrete and continuous environments, with six requiring discrete actions and the other 14 continuous ones. The number of actions can range from a single action to 19 different actions.

**Quality of Reward** We cover different types of reward signals with our benchmarks, ranging from relatively sparse step penalty style rewards where the agent only receives a reward of $-1$ each step to complex composite reward functions in e.g. the Brax-based environments. The latter type is quite informative, providing updates on factors like movement economy and progress toward the goal whereas the former does not let the agents distinguish between transitions without looking at the whole episode. Further examples for sparse rewards are the CARLCartPoleEnv and CARLVehicleRacingEnv.

**Context Spaces** While the full details of all possible context configurations can be seen in Appendix H, for brevity here we only discuss the differences between context spaces and the configuration possibilities they provide. Depending on the environment the context features have different influences on the dynamics and the reward. Of all 145 registered context features, 99 % influence the dynamics. This means that if a context feature is changed then the transition from states to their successors is affected and likely changed as well. Only 4 % of the context features shape the reward. Most context features (91 %) are continuous; the rest are categorical or discrete.

**Summary** Comparing our benchmarks along these attributes, we see a wide spread in most of them (Figure 5). `CARL` focuses on popular environments and will grow over time, increasing the diversity of benchmarks. Already now, `CARL` provides a benchmarking collection that tasks agents with generalizing in addition to solving the problem most common in modern RL while providing a platform for reproducible research.

## 6 Experiments

Having discussed the framework of cRL and the implications of context in training, we now study several research questions regarding the empirical effects of context: (i) How much does varying context influence performance? Can agents compensate across context variations in a zero-shot manner? (ii) Can we observe the effects discussed in Section 4 in practice? I.e., is there an observable optimality gap on cMDPs, does the context visibility influence the performance and which role does the width of the context distribution play? (iii) How can we assess generalization performance, and how does the test behavior of agents change when exposed to the context information?

To explore our research questions, we use our benchmark library `CARL`. Details about the hyperparameter settings and used hardware for all experiments are listed in Appendix C. In each experiment, if not specified otherwise, we train and evaluate on 10 different random seeds and a set of 128 uniformly sampled contexts. All experiments can be reproduced using the scripts we provide with the benchmark library at https://github.com/automl/CARL.

### 6.1 How Does Varying Context Influence Performance?

To get an initial understanding on the generalization capabilities, we train a well-known SAC agent (Haarnoja et al., 2018) on the default version of the Pendulum (Brockman et al., 2016) environment. Pendulum is a very simple environment (see Appendix B for dynamic equations) compared to the majority of RL benchmarks and has been considered solved by deep RL for years. However, we show that we can increase the difficulty of this environment substantially when considering even single context features. Note that this increase in difficulty also means the best return may decrease even for an optimal agent, in extreme cases making instances impossible to solve. The agent is not provided with any explicit information about the context, i.e., it is context-oblivious. Then, for evaluation, we vary each defined context feature by magnitudes $A = 0.1, 0.2, \ldots 2.2$ times the default value for 10 test episodes. In Figure 6, we plot the empirical cumulative distribution functions (eCDF) for the return showing the range of observed returns on the x-axis and the proportion on the y-axis. The further to the right the curve is, the better the observed returns. For the eCDF plots of other `CARL` environments, see Appendix D.1.

First and foremost we observe that some context features do not have an influence on the generalization performance when varied, see Figure 6. Even in this zero-shot setting, the agent performs similarly well on all context variations of the `initial_angle_max` and `initial_velocity_max` features. Yet, the agent's performance is very brittle to other context features, i.e. `max_speed`, simulation timestep `dt`, gravity `g` and length `l`. The trained agent cannot compensate for the effect the context has on the policy and thus performs poorly on several context variations. It is worth noting that the performance curves across the variations depends on the context feature. `max_speed`, for example, is hard for values below $A = 0.7$, but then the challenge level for the agent decreases abruptly. This transition is smoother for `l` where both very small and very large values are hard to solve. We conclude that it is not straightforward to estimate the impact of changing the environment on agent behavior, especially for physics simulations or similarly complex environments. We also clearly see that zero-shot generalization cannot compensate for variations in environment dynamics. Context variations introduce a significant challenge, even on a simple environment like Pendulum. The next step, therefore, is to train the agent on varying contexts and re-evaluate its generalization performance.

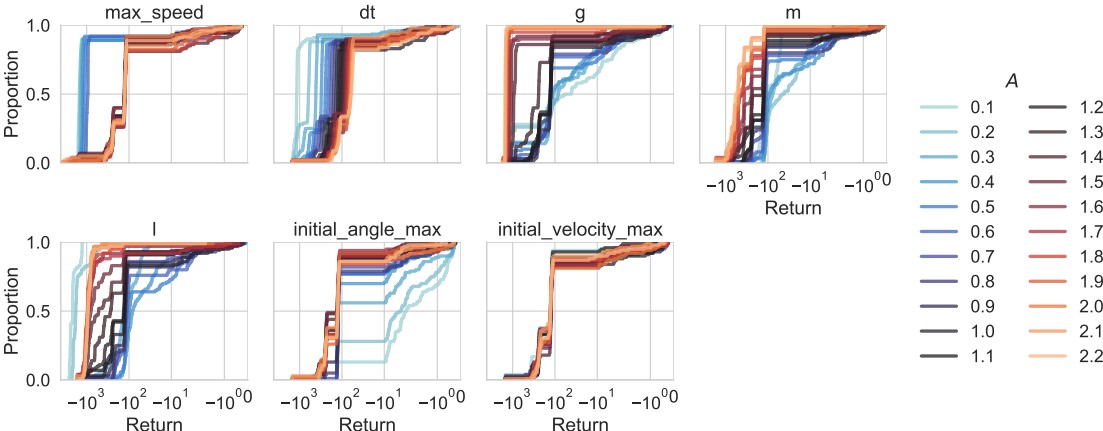

Figure 6: **CARLPendulumEnv**: eCDF Plot. $A$ is the magnitude multiplied with the default value of each context feature. So, $A = 1.0$ refers to the standard environment.

### 6.2 Does the Optimality Gap Exist in Practice?

In the previous section, we saw that context variation heavily influences the test performance of a standard agent. Here, we take a closer look and connect this to the Optimality Gap $\mathcal{OG}$ (Equation (1)). In order to demonstrate how significant this gap is in practice, we train a C51 agent (Bellemare et al., 2017) on our contextually extended CartPole environment Brockman et al. (2016) as well as a SAC agent (Haarnoja et al.,

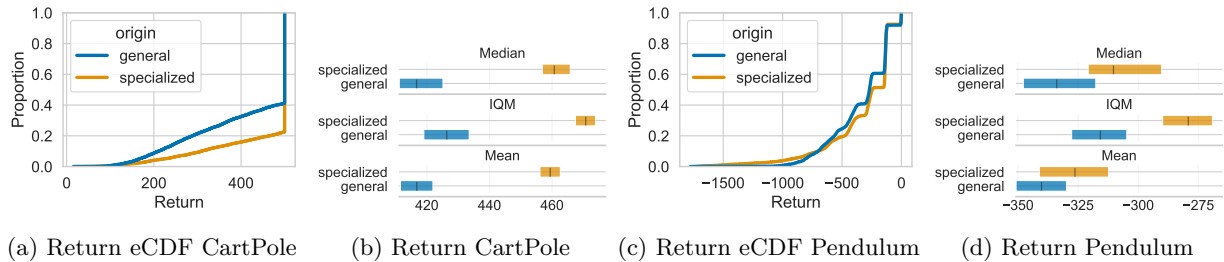

(a) Return eCDF CartPole     (b) Return CartPole     (c) Return eCDF Pendulum     (d) Return Pendulum

Figure 7: Optimality Gap on CARLCartPole (left) and CARLPendulum (right).

2018) on Pendulum. Similarly as above, we use simple environments to demonstrate the induced difficulty by context variation.

To generate instances of both environments, we vary the length of the pole across a uniform distribution $p_{\mathcal{C}} = \mathcal{U}(0.25, 0.75)$ around the standard pole length for CartPole and the pole length across $p_{\mathcal{C}} = \mathcal{U}(1, 2.2)$ for Pendulum. For training, we sample 64 contexts from this distribution and train a general agent which experiences all contexts during training in a round robin fashion. However, we do not explicitly provide the context information to the agent. We approximate the optimal performance by training a separate specialized agent on each context. Afterwards, each agent is evaluated on each context it was trained on for 10 episodes.

Comparing the general and specialized agents, we see a difference of at least 30 reward points in median, mean and estimated IQM performance (as proposed by Agarwal et al. (2021)) for CartPole and a smaller, but similar effect on Pendulum. While this is not a huge decrease in performance, looking at how these rewards are distributed across the evaluation instances shows that the general agent solves significantly fewer instances than the specialized agent with a decrease of around 40% of finished episodes on CartPole. This shows that while most instances can be solved by the agent one at a time, training an agent that solves all of them jointly is a significant challenge.

### 6.3 Does Access To Context Improve Training?

We have seen that agents without access to context information are not always able to entirely solve even simple contextual environments. Can these agents improve with access to the context information? We choose a very simple approach of adding either the whole context (concat all) or only the actively changing context feature (concat non-static) to the observation. Obviously this is a simplistic approach that in the case of concat all significantly alters the size of the observation. Though, even with this simple idea, training performance improves by a large margin in some cases, see Figure 8. Here, the agent, this time on CARLDmcWalker with changing viscosity ($\sim \mathcal{U}[1, 2.5]$ times the default value), learns faster, is more stable, and reaches a higher final performance with the additional information added to the observation. In testing, we see significantly more failures in the hidden agent compared to the concat ones, with only the concat non-static agent learning to solve the contextual environment without a large decrease in overall test performance. Effective generalization seems to be a matter of a reasonable feature set, similar to supervised learning.

On Pendulum, however, this is not the case; we see no meaningful difference in mean performance of concat (non-static) and hidden when varying length ($\sim \mathcal{U}[1, 2.2]$ times the default value). Please note that the large confidence interval stems from runs where the algorithm did not find a meaningful performance, see Appendix Figure 11. The concat agents perform better on some unseen contexts in evaluation, though the hidden agent is far superior on a slice of the instance set.

In both cases, the generalization behavior of contextual agents is different from the hidden one. Most `CARL` environments show that additional context information is indeed beneficial for train and test performance as in Walker (see Appendix E for full results). Additionally, the performances of concat all and concat non-static agents provide no clear pattern as to how many context features should be provided in training. We conclude that while context information is indeed useful for generalization performance, simply appending the context to the observation might not be the ideal way to communicate context features. Instead, context embeddings could be a more potent way of capturing the way a context feature changes an environment,

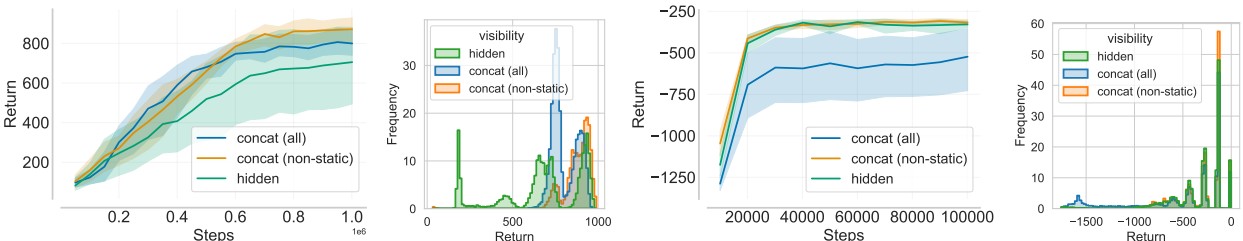

Figure 8: Train (lineplot) and test performance (histogram) of agents with visible and hidden context on CARLDmcWalker with different viscosity values and 5 seeds (left) and CARLPendulum with different lengths and 20 seeds (right). Shown is the mean performance with 95% confidence interval and testing across 200 test contexts (metrics are computed using stratified resampled bootstrapping (Agarwal et al., 2021)).

as we have seen in some prior work on incorporating goals (Sukhbaatar et al., 2018; Liu et al., 2022) into training. Since our goal was to show the potential of context information, we leave it to future work to investigate better representations of context features.

### 6.4 How Far Can Agents Learn To Act Across Contexts?

As we saw different evaluation behaviors from hidden and visible agents in the last section, we want to further investigate their generalization capabilities in- and out-of-distribution. To this end, we follow a three mode evaluation protocol for Contextual Reinforcement Learning that tests the agent's interpolation capabilities and out-of-distribution generalization under different training distribution shapes (Kirk et al., 2023). This is in contrast to PCG environments where we cannot define evaluation protocols and instead have to rely on the given instance generation procedure.

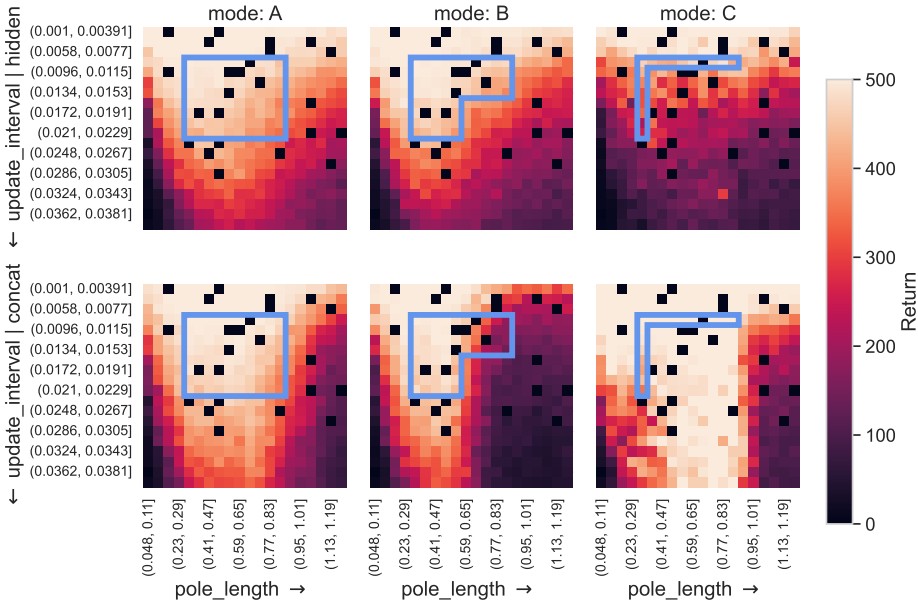

Figure 9: In- and Out-of-Distribution Generalization on CartPole. We vary the pole length and update_interval. The blue polygon marks the train context area. Black dots mark gaps in the context space due to random sampling. First row: Context-oblivious agent, second row: Concat.

We define train and test distributions for each dimension of the context space individually, allowing us to study different relationships between train and test settings. If at least parts of the test context are within the train distribution, we speak of interpolation, if the whole context is outside, of extrapolation. By choosing two context features and defining uniform train distributions on both, we construct a convex training set in

the context feature space (mode A: □). The context feature distributions can also be defined to allow only a small variation (mode B: ⌐) or a single value per feature (mode C: ⌐), creating non-convex train sets. Thus, the convex hull of this non-convex set tests combinatorial interpolation.

To demonstrate this, we again choose contextual CartPole from `CARL`, and the C51 (Bellemare et al., 2017) agent known to perform well on it. We train the agent for 100 000 timesteps and vary the update_interval and the pole length in the environment, once without access to the context (hidden) and once concatenating pole length and gravity to the observation (concat). We repeat this with 10 random seeds and 5 test episodes per context. For the train and test context sets, we sample 1000 contexts each for the train and test distributions defined in the evaluation protocol, see Figure 9. The test performances are discretized and aggregated across seeds by the bootstrapped mean using rliable (Agarwal et al., 2021).

In Figure 9, we show that both hidden (context-oblivious) and visible (concatenate) agents perform fairly well within their training distribution for evaluation mode A and even generalize to fairly large areas of the test distribution, more so for concat. Large update intervals combined with extreme pole lengths proves to be the most challenging area.

Interestingly, the concat agent is able to solve more of the large update intervals in contrast to the hidden agent which is most pronounced on train distribution C. This itself is counterintuitive, we would expect that the larger the train distribution, the larger the out-of-distribution generalization. The hidden agent in general performs well for low update intervals. This resonates with the intuition that smaller update intervals are easier because there is more granularity (and time) to react to the current state. In addition, we provide results for varying the update interval and the gravity. Here, the results are similar but subdued, see Appendix Figure 12. Varying the pole length together with the gravity paints a different image (Appendix Figure 13). In this case, the hidden agent performs much better. We suspect that the effects of gravity and pole length cancel out and thus context information is not needed to learn a meaningful policy. These three variations again show that providing context by concatenation *can* be helpful but not in every case, demanding further investigation on alternatives. Finally, neither agent shows reliable combinatorial interpolation performance, let alone out-of-distribution generalization. We see here a major open challenge for the RL community, for which `CARL` will support them in the development and precise studies of RL generalization capabilities.

## 7    Conclusion

Towards our goal of creating general and robust agents, we need to factor in possible changes in the environment. We propose modeling these changes with the framework of contextual Reinforcement Learning (cRL) in order to better reason about what demands Contextual Reinforcement Learning introduces to the agents and the learning process, specifically regarding the suboptimal nature of conventional RL policies in cRL. With `CARL`, we provide a benchmark library which contextualizes popular benchmarks and is designed to study generalization in Contextual Reinforcement Learning. It allows us to empirically demonstrate that contextual changes disturb learning even in simple settings and that the final performance and the difficulty correlate with the magnitude of the variation. We also verify that context-oblivious policies are not able to fully solve even simple contextual environments. Furthermore, our results suggest that exposing the context to agents even in a naive manner impacts the generalization behavior, in some cases improving training and test performance compared to non-context-aware agents. We expect this to be a first step towards better solution mechanisms for contextual RL problems and therefore one step closer to general and robust agents.

### Broader Impact Statement

We foresee no new direct societal and ethical implications other than the known concerns regarding autonomous agents and RL (e.g., in a military context).

### Acknowledgments

The work of Frederik Schubert, Sebastian Döhler and Bodo Rosenhahn was supported by the Federal Ministry of Education and Research (BMBF), Germany under the project LeibnizKILabor (grant no. 01DD20003) and the AI service center KISSKI (grant no. 01IS22093C), the Center for Digital Innovations (ZDIN) and the Deutsche Forschungsgemeinschaft (DFG) under Germany's Excellence Strategy within the Cluster of

Excellence PhoenixD (EXC 2122). André Biedenkapp and Frank Hutter acknowledge funding through the research network "Responsive and Scalable Learning for Robots Assisting Humans" (ReScaLe) of the University of Freiburg. The ReScaLe project is funded by the Carl Zeiss Foundation.

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

# Appendix

## A    Partial Observability in cMDPs

Discussing the visibility of context for an agent can be linked to the partial observability we see in POMDPs. We believe it is useful to differentiate between the visibility of context and state features or observations as both serve a different function in a cMDP. The state features describe the current state while the context describes the current MDP. Therefore making one only partially observable should influence the learning dynamics in different ways. Therefore we define a cMDP as a special case of a POMDP, analogous to Kirk et al. (2023), where we have an emission function $\phi : \mathcal{S} \times p_{\mathcal{C}} \to O_s \times O_c$ mapping the state space to some state observation space $O_s$ and context observation space $O_c$. $\phi$ differentiates between state $s$ and context $c$ to allow different degrees of observability in state and context, e.g. hiding the context completely but exposing the whole state, in order to enable more flexible learning. It can also introduce the additional challenge of learning from imperfect or noisy context information.

## B    Pendulum's Dynamic Equations

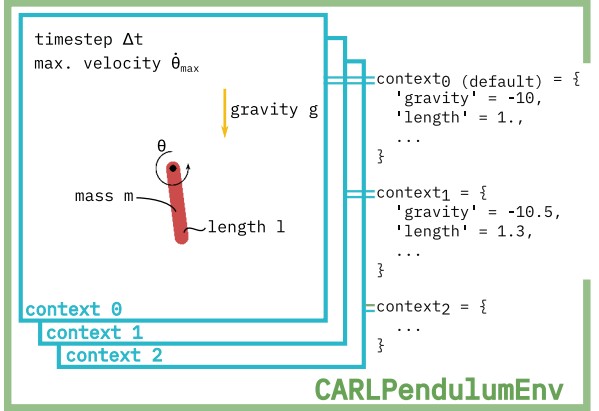

Figure 10: CARLPendulumEnv

Because we use CARLPendulumEnv embedding gym's Pendulum (Brockman et al., 2016) for our task variation experiment (see Section 6.1), we provide the dynamic equations to show the simplicity of the system. The state and observation consists of the angular position $\theta$ and velocity $\dot{\theta}$ of the pendulum. The discrete equation defining the behavior of the environment is defined as follows:

$$\dot{\theta}_{k+1} = \dot{\theta}_k + \left( -\frac{3g}{2l} \sin(\theta_k + \pi) + \frac{3}{m \cdot l^2} u_k \right) \cdot \Delta t$$

$$\theta_{k+1} = \theta_k + \dot{\theta}_{k+1} \cdot \Delta t \,.$$

Here, $k$ is the index of the iteration/step. The dynamic system is parametrized by the context, which consists of $g$ the gravity, $l$ and $m$ the length and mass of the pendulum, $u$ the control input and $\Delta t$ the timestep. Figure 10 shows how Pendulum is embedded in CARL.

## C    Hyperparameters and Hardware

**Hyperparameters and Training Details**   We implemented our own agents using coax (Holsheimer et al., 2023) with hyperparameters specified in Table 1. All experiments can be reproduced using the scripts we provide with the benchmark library at `https://anonymous.4open.science/r/CARL-54F4/`.

**Hardware**   All experiments on all benchmarks were conducted on a slurm CPU and GPU cluster (see Table 2). On the CPU partition there are 1592 CPUs available across nodes.

## D    Additional Experimental Results

In this section, we provide additional information and results for our experiments section (section 6).

### D.1    Task Variation Through Context

Following the experimental setup in Section 6.1 we conducted further experiments on representative CARL-environments.

Table 1: Hyperparameters for algorithm and environment combinations

| algorithm | c51 | c51 | sac | c51 | c51 | sac | sac | sac |
|---|---|---|---|---|---|---|---|---|
| env | CartPole | Acrobot | Pendulum | MountainCar | LunarLander | DmcWalker | DmcQuadruped | Halfcheetah |
| n_step | 5 | 5 | 5 | 5 | 5 | 5 | 5 | 5 |
| gamma | 0.99 | 0.99 | 0.9 | 0.99 | 0.99 | 0.9 | 0.9 | 0.99 |
| alpha | 0.2 | 0.2 | 0.2 | 0.2 | 0.2 | 0.2 | 0.2 | 0.1 |
| batch_size | 128 | 128 | 128 | 128 | 128 | 128 | 128 | 256 |
| learning_rate | 0.001 | 0.001 | 0.001 | 0.001 | 0.001 | 0.001 | 0.001 | 0.0001 |
| q_targ_tau | 0.001 | 0.001 | 0.001 | 0.001 | 0.001 | 0.001 | 0.001 | 0.005 |
| warmup_num_frames | 5000 | 5000 | 5000 | 5000 | 5000 | 5000 | 5000 | 5000 |
| pi_warmup_num_frames | 7500 | 7500 | 7500 | 7500 | 7500 | 7500 | 7500 | 7500 |
| pi_update_freq | 4 | 4 | 4 | 4 | 4 | 4 | 4 | 2 |
| replay_capacity | 100000 | 100000 | 100000 | 100000 | 100000 | 100000 | 100000 | 1000000 |
| network | {'width': 256, 'num_atoms': 51} | {'width': 256, 'num_atoms': 51} | {'width': 256} | {'width': 32, 'num_atoms': 51} | {'width': 256, 'num_atoms': 51} | {'width': 256} | {'width': 256} | {'width': 1024} |
| pi_temperature | 0.1 | 0.1 | NaN | 0.1 | 0.1 | NaN | NaN | NaN |
| q_min_value | 0.0 | -50.0 | NaN | -100.0 | -100.0 | NaN | NaN | NaN |
| q_max_value | 110.0 | 0.0 | NaN | 100.0 | 100.0 | NaN | NaN | NaN |

Table 2: GPU cluster used for training

| Type | Model | Quantity | RAM CPU (G) |
|---|---|---|---|
| GPU | NVIDIA Quattro M5000 | 1 | 256 |
| GPU | NVIDIA RTX 2080 Ti | 56 | 384 |
| GPU | NVIDIA RTX 2080 Ti | 12 | 256 |
| GPU | NVIDIA RTX 1080 Ti | 6 | 512 |
| GPU | NVIDIA GTX Titan X | 4 | 128 |
| GPU | NVIDIA GT 640 | 1 | 32 |

## D.2 Adding Context to the State

When we concatenat all available context features to the observation for CARLPendulum, we often see that the algorithm fails to learn a meaningful policy on some seeds, see Figure 11.

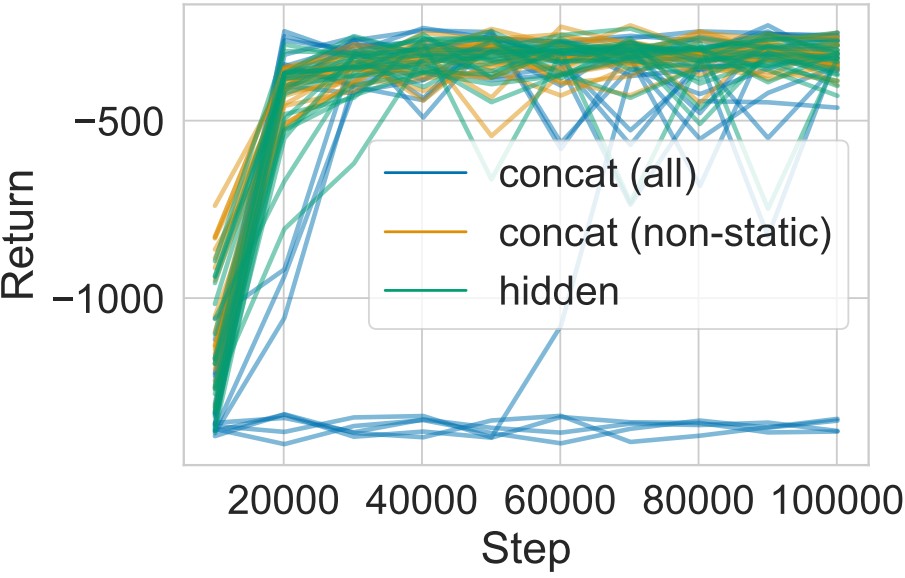

Figure 11: CARLPendulum with different lengths and 20 seeds. Train performance.

### D.3 Generalization Results

Here we provide two more combinations for CARLPendulum for the Kirk generalization protocol (Kirk et al., 2021) from Section 6.4 (same experimental setup).

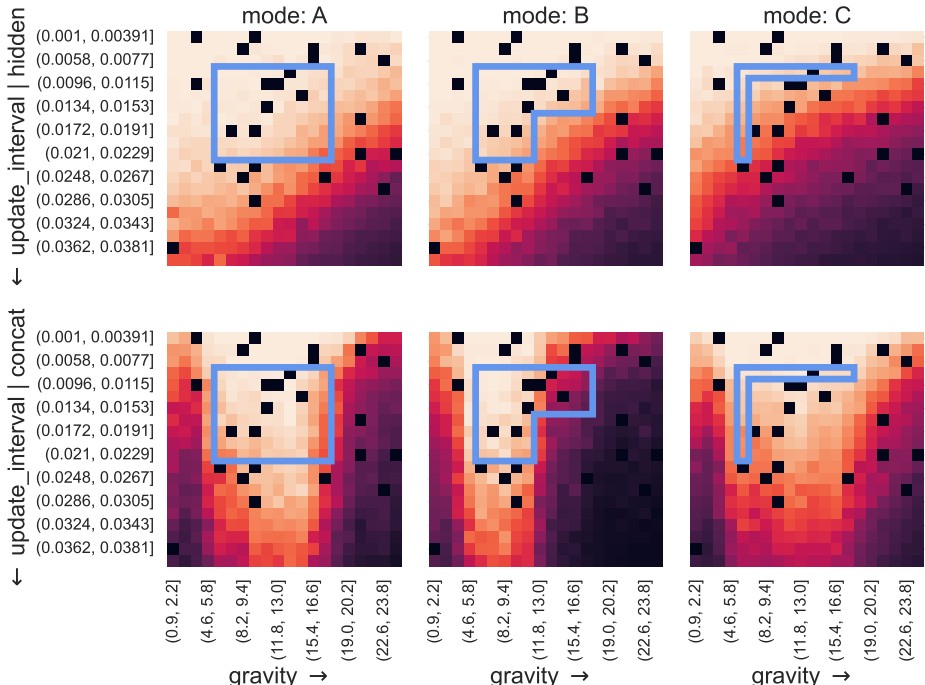

Figure 12: Varying gravity and update interval on CARLPendulumEnv. First row: Hidden agent, second row: Concat agent.

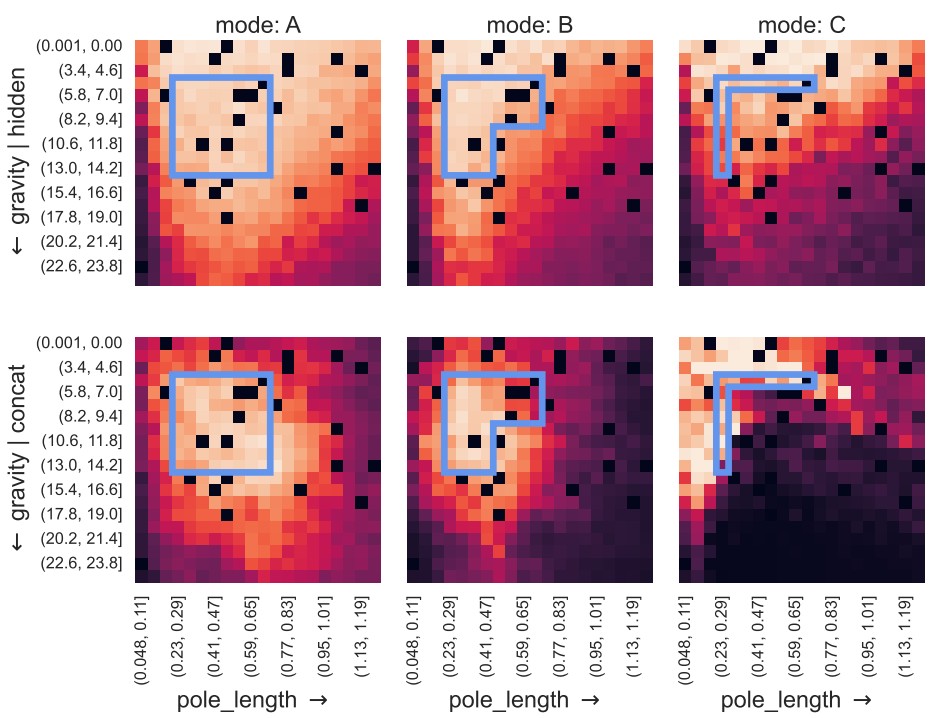

Figure 13: Varying gravity and pole length on CARLPendulumEnv. First row: Hidden agent, second row: Concat agent.

# E   Baselines

Here we provide baselines for selected environments in CARL. For each environment we conduct the following experiments. First, we train a default agent on the default environment, i.e. with no context variation. This agent is then evaluated on context variations of single context features with a magnitude $A \in \{0.1, 0.2, \ldots, 2.2\}$. This creates an initial performance profile and shows how sensitive the agent is to which context feature. We plot this as an eCDF, the further right the lines are, the more high returns the agent achieves. This plot features no aggregation. After that, we train an agent *with* context variation. For this we select a context feature with a visible impact on performance and determine the range by including magnitudes where the environment is still solvable which means reaching returns better than fail. We train the agent with varying context visibility: Once, the varying context feature is hidden, once the complete context set is concatenated to the state and once only the changing context feature. We plot the training curve. Finally, we test the agents trained on context variation on the same number of contexts sampled from the train distribution and report the return histogram and return statistics. If not specified otherwise, we perform each experiment with 10 seeds and 128 training contexts.

We report on classic control (CARLPendulumEnv: Figure 14, CARLMountainCarEnv: Figure 15, CARL-CartPoleEnv: Figure 16 and CARLAcrobotEnv: Figure 17) and box2d (CARLLunarLanderEnv: Figure 18).

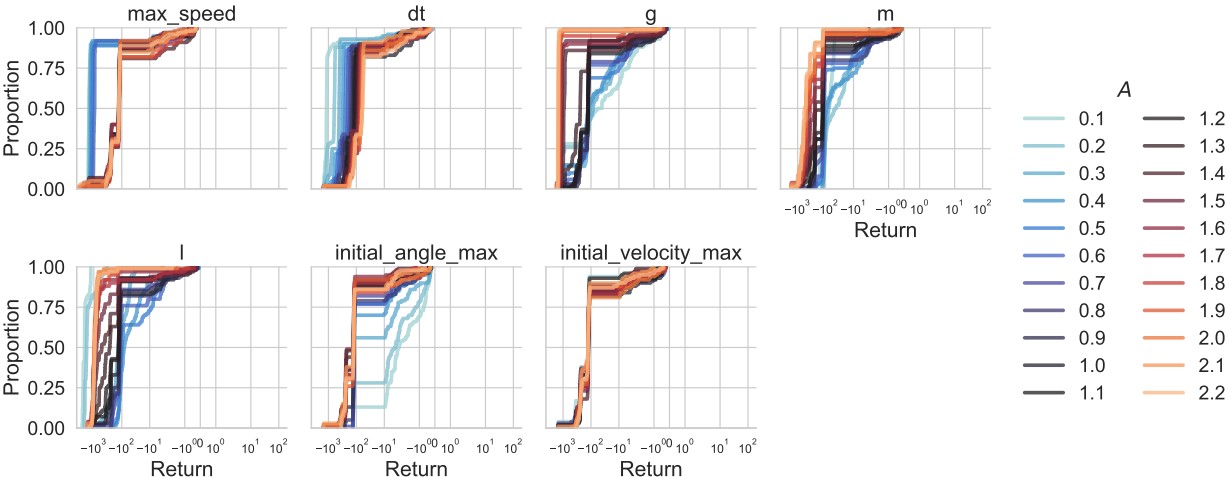

(a) Default agent evaluated on context variation. eCDF Plot. A is the magnitude multiplied with the default value of each context feature.

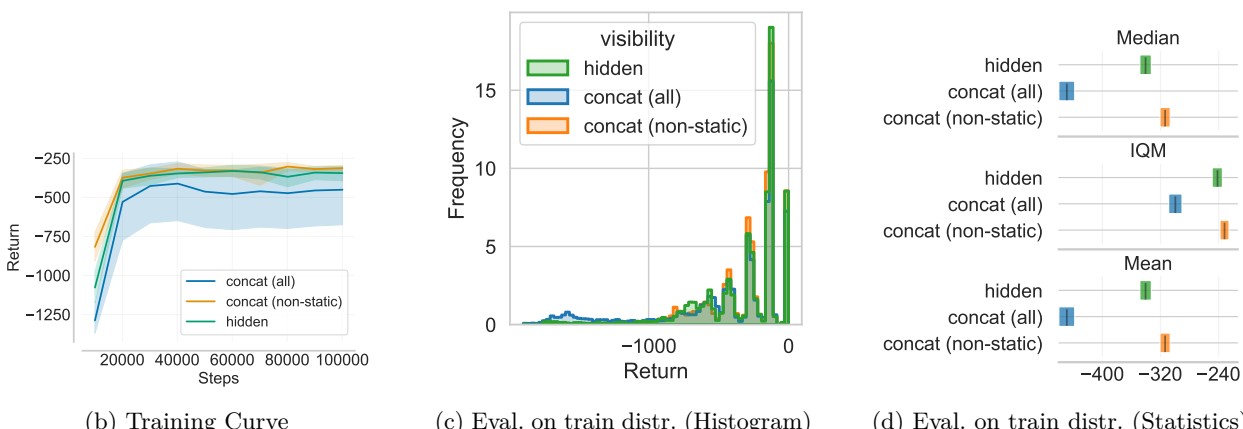

(b) Training Curve    (c) Eval. on train distr. (Histogram)    (d) Eval. on train distr. (Statistics)

Figure 14: **CARLPendulumEnv**: Benchmark. Algorithm sac, 10 seeds. (a): Default agent (trained on default context and context-oblivious) evaluated on context variations. (b-d): Training with varying context visibility. We vary the context feature(s) ['l'] ($A \sim \mathcal{U}(0.5, 2.2)$).

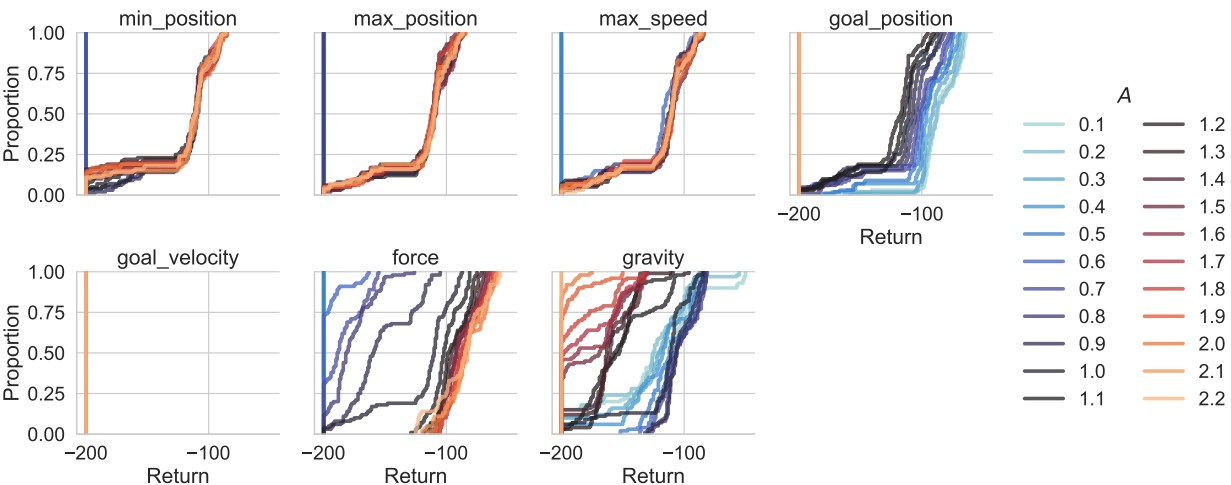

(a) Default agent evaluated on context variation. eCDF Plot. A is the magnitude multiplied with the default value of each context feature.

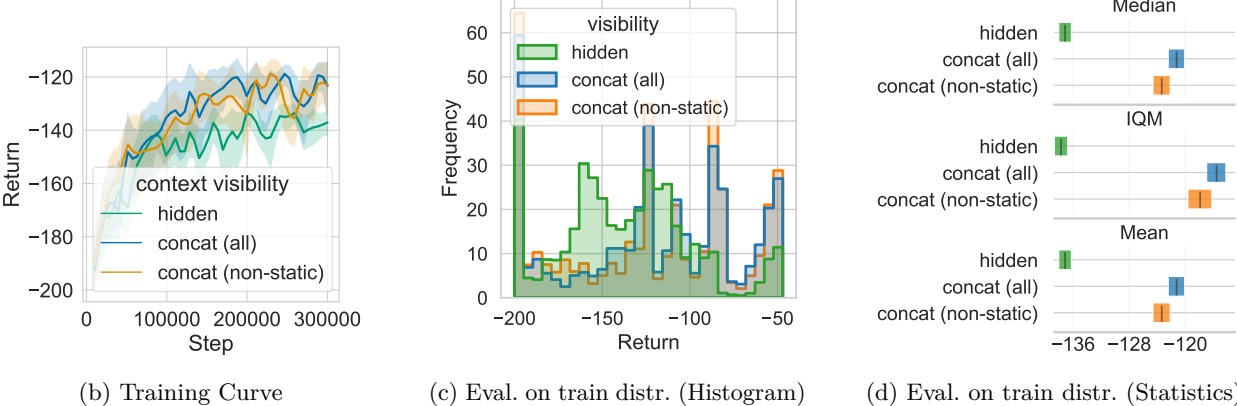

(b) Training Curve      (c) Eval. on train distr. (Histogram)      (d) Eval. on train distr. (Statistics)

Figure 15: **CARLMountainCarEnv**: Benchmark. Algorithm c51, 10 seeds. (a): Default agent (trained on default context and context-oblivious) evaluated on context variations. (b-d): Training with varying context visibility. We vary the context feature(s) ['gravity'] $(A \sim \mathcal{U}(0.1, 2))$.

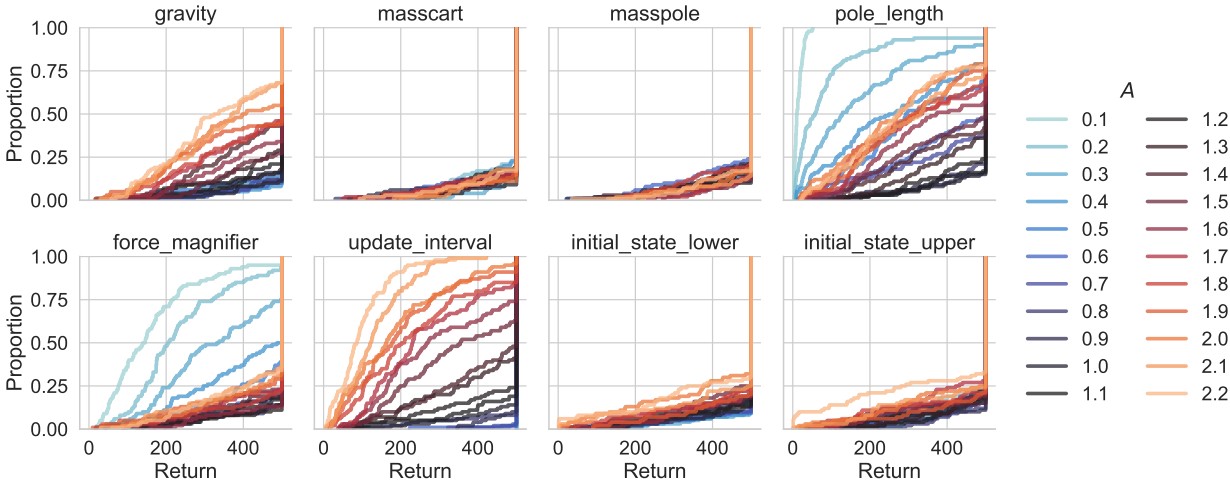

(a) Default agent evaluated on context variation. eCDF Plot. A is the magnitude multiplied with the default value of each context feature.

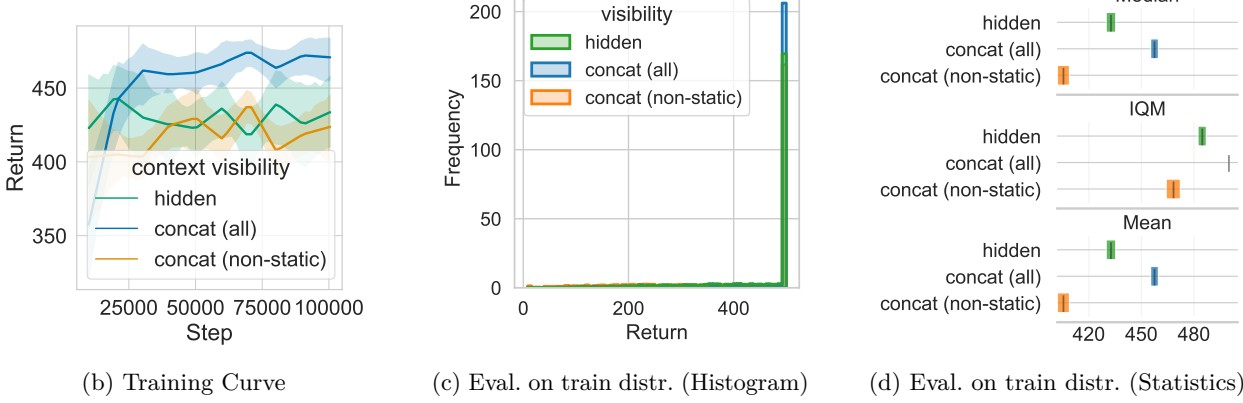

(b) Training Curve      (c) Eval. on train distr. (Histogram)      (d) Eval. on train distr. (Statistics)

Figure 16: **CARLCartPoleEnv**: Benchmark. Algorithm c51, 10 seeds. (a): Default agent (trained on default context and context-oblivious) evaluated on context variations. (b-d): Training with varying context visibility. We vary the context feature(s) ['pole_length'] ($A \sim \mathcal{U}(0.2, 2.2)$).

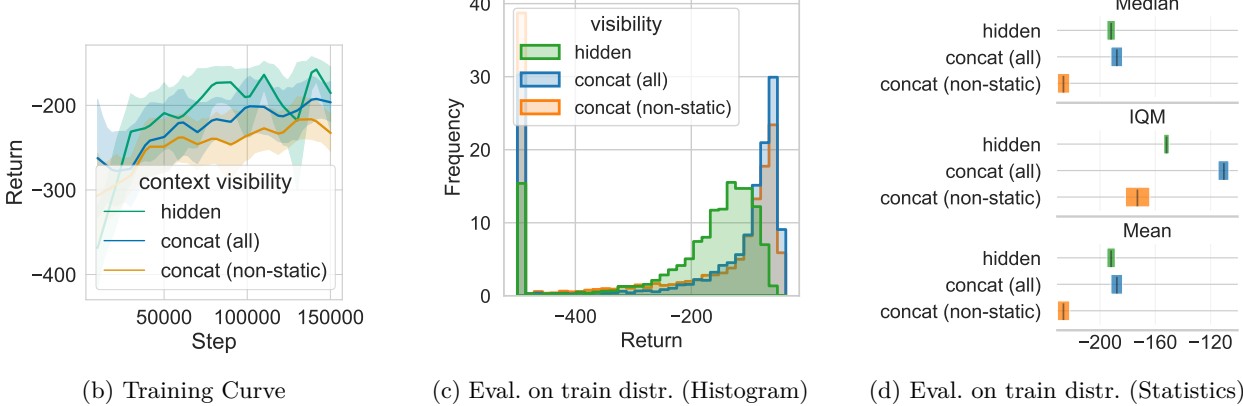

Figure 17: **CARLAcrobotEnv**: Benchmark. Algorithm c51, 10 seeds. (a): Default agent (trained on default context and context-oblivious) evaluated on context variations. (b-d): Training with varying context visibility. We vary the context feature(s) `['link_mass_2']` ($A \sim \mathcal{U}(0.1, 2.2)$).

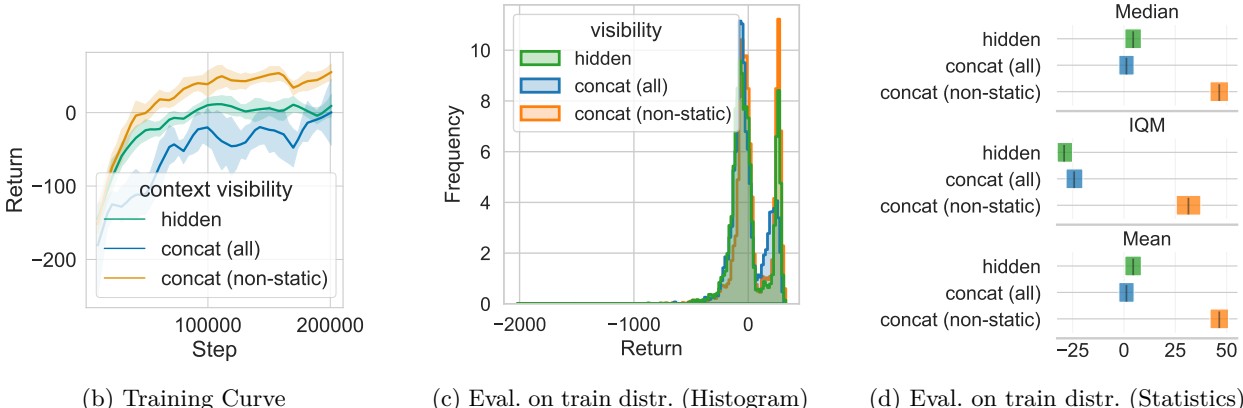

(a) Default agent evaluated on context variation. eCDF Plot. A is the magnitude multiplied with the default value of each context feature.

(b) Training Curve    (c) Eval. on train distr. (Histogram)    (d) Eval. on train distr. (Statistics)

Figure 18: **CARLLunarLanderEnv**: Benchmark. Algorithm c51, 10 seeds. (a): Default agent (trained on default context and context-oblivious) evaluated on context variations. (b-d): Training with varying context visibility. We vary the context feature(s) ['GRAVITY_Y'] ($A \sim \mathcal{U}(0.1, 2.2)$).

## E.1 Context-Conditioning on CARLMarioEnv

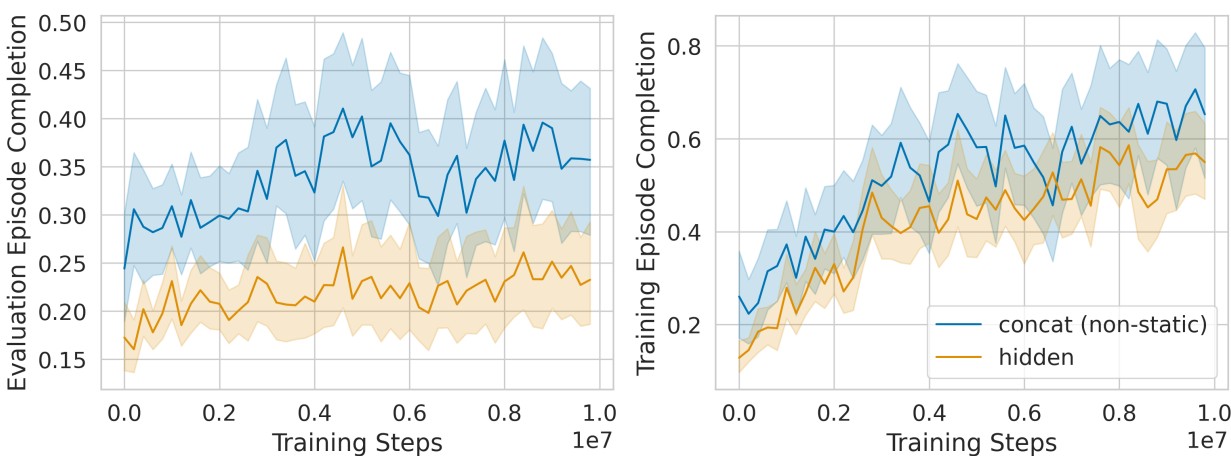

Figure 19: **CARLMarioEnv**: Evaluation of the PPO agent on 16 TOAD-GAN training levels and 16 different test levels. The episode completion indicates the average distance from the level start that the agent is able to reach.

We train the Proximal Policy Optimization (PPO) agent on 16 distinct levels of the CARLMarioEnv environment and evaluate its performance on another 16 different levels for 10 seeds. In the context-aware variant of the agent, both the policy and value functions are conditioned on the provided context. Unlike the context-aware agent, the hidden agent only receives RGB frames from the environment as input. The context is encoded using a convolutional encoder and integrated into the hidden state representation, which is then fed into the policy and value heads. The context itself consists of a noise tensor sampled from which TOAD-GAN (Awiszus et al., 2020; Schubert et al., 2021) generated the Super Mario Bros. levels. To ensure playability, the generated levels are filtered using static analysis.

As shown in Figure 19, the training performance of the context-aware agent and the agent without context is nearly identical. However, when evaluated, the context-aware agent outperforms the agent without context, demonstrating its ability to effectively incorporate the noise map context into its policy. This observation underscores the value of the CARL benchmark in driving research on context representation.

## E.2 Test Performance on Context Variations

In this section we show the test performance of an agent trained on the default context of additional selected environments which is oblivious to the context. For evaluation we run 10 episodes on contexts with different magnitudes of variation. We vary each context feature by a magnitude $A = 0.1, 0.2, 0.3, \ldots, 2.2$.

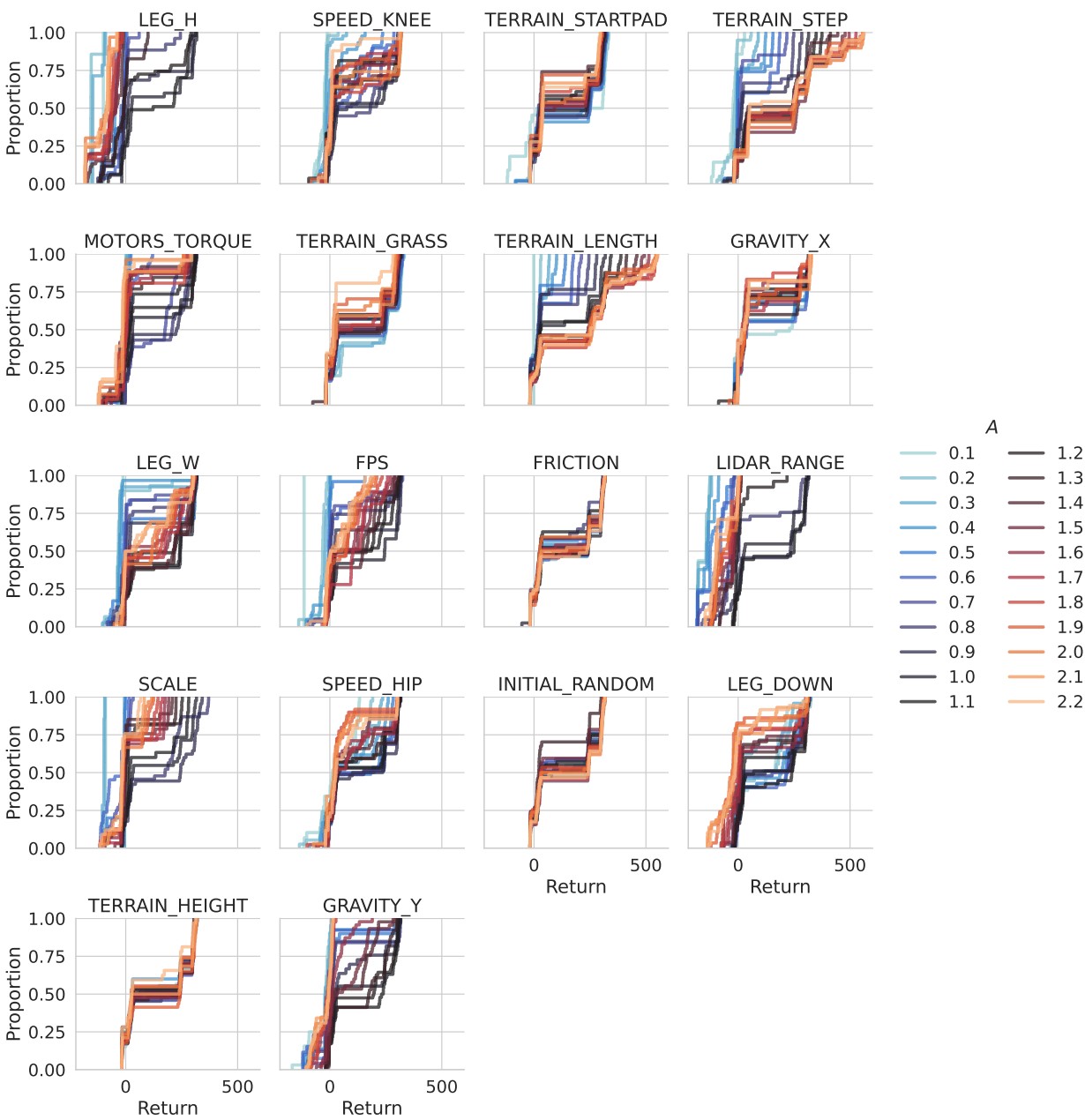

Figure 20: **CARLBipedalWalkerEnv**: ECDF Plot. A is the magnitude multiplied with the default value of each context feature.

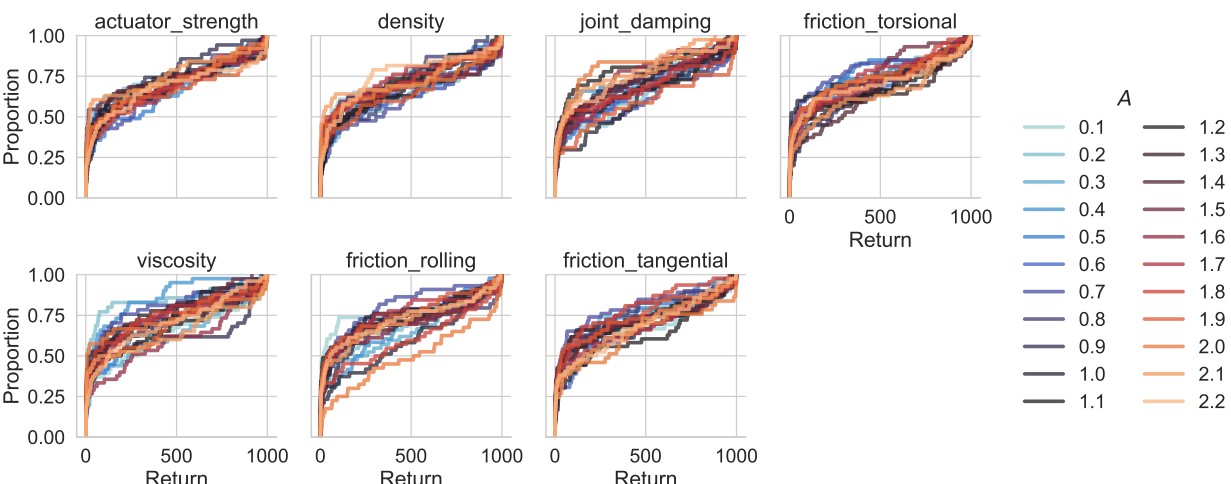

Figure 21: **CARLDmcFishEnv**: ECDF Plot. A is the magnitude multiplied with the default value of each context feature.

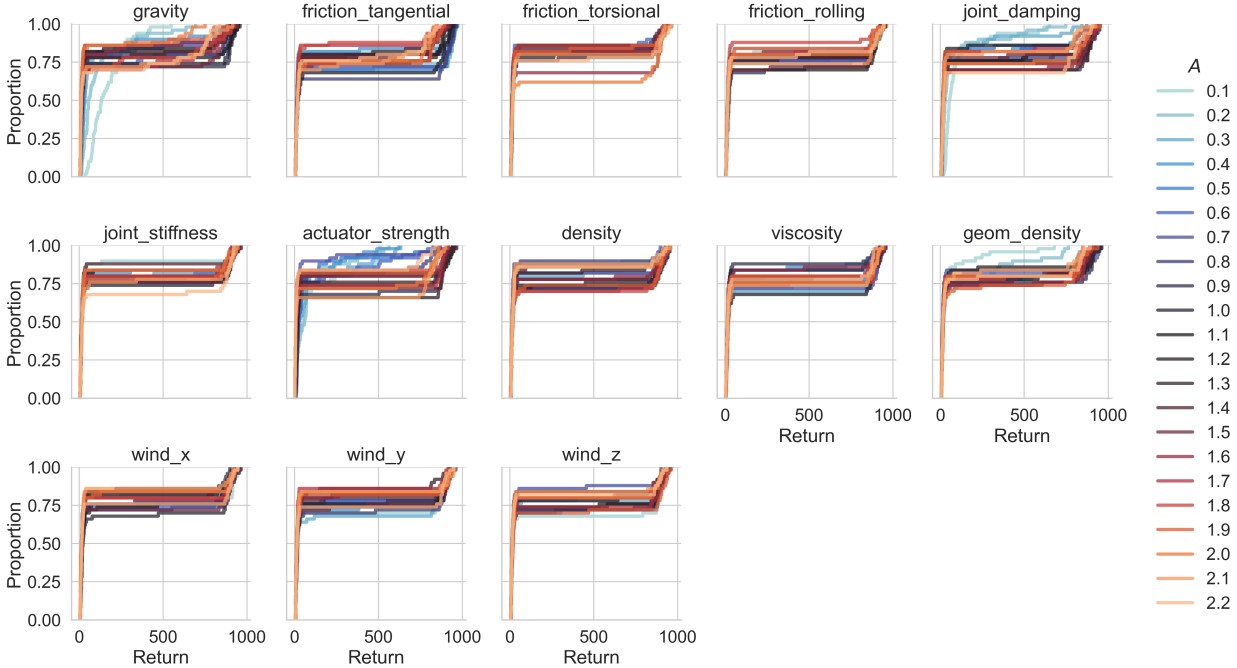

Figure 22: **CARLDmcQuadrupedEnv**: ECDF Plot. A is the magnitude multiplied with the default value of each context feature.

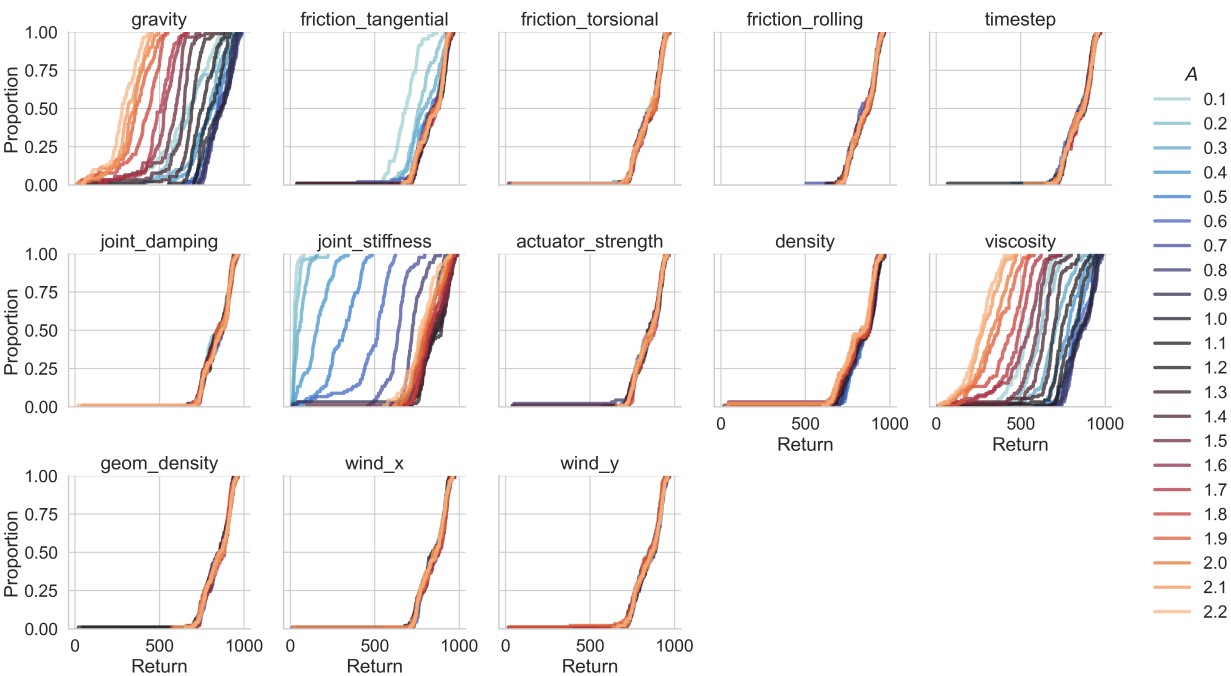

Figure 23: **CARLDmcWalkerEnv**: ECDF Plot. A is the magnitude multiplied with the default value of each context feature.

## F  Hyperparameter Optimization in cRL

We have observed significant differences in learning performance for hidden and full visible context, but the same is also true for hyperparameter tuning in both of these settings. We use the same DQN and DDPG algorithms as in our other experiments with a narrow context distribution of 0.1 for the `CARL` Pendulum, Acrobot and LunarLander environments to show this point. To tune the hyperparameters, we use PB2 (J. Parker-Holder et al., 2020) for the learning rate, target update interval and discount factor.

As shown in Figure 24, the evaluation performances of the found hyperparameter schedules differ significantly in terms of learning speed, stability and results *per environment*. Providing the context sometimes seems to increase the difficulty of the problem (see Section 6.4), i.e. finding a good hyperparameter configuration happens more often and more reliably when the policy is not given the context. We can only speculate on the reasons why this happens, but shows that context introduces complexities to the whole training process beyond simply the policy architecture.

## G  Open Challenges in cRL

We used the concept of Contextual Reinforcement Learning and its instantiation in `CARL` to demonstrate the usefulness of context information in theory and in practice. More specifically, we showed that making such information about the environment explicitly available to the agent enables faster training and transfer of agents (see Section 6). While this already provides valuable insights to the community that increasingly cares about learning agents capable of generalization (see Sections 1 & 3) Contextual Reinforcement Learning and by extension `CARL` enables to study further open challenges for general RL.

### G.1  Challenge I: Representation Learning

Our experiments demonstrated that an agent with access to context information can be capable of learning better than an agent that has to learn behaviors given an implicit context via state observations, but the naive method of including context information in the observation is not reliable. We theorize that disentangling the representation learning aspect from the policy learning problem reduces complexity. As `CARL` provides ground truth for representations of environment properties we envision future work on principled studies of novel RL algorithms that, by design, disentangle representation learning and policy learning (see, e.g., (Rakelly et al., 2019; Fu et al., 2021a; Zhang et al., 2021b) as first works along this line of research). The ground truth given by the context would allow us to measure the quality of learned representations and allows us to relate this to the true physical properties of an environment.

Another direction of research under the umbrella of representation learning follows the work of environment probing policies (Zhou et al., 2019). There, exploratory policies are learned that allow one to identify which environment type an agent encounters. This is complementary to the prior approaches as representations are not jointly learned with the behaviour policies as in the previously discussed approaches but rather in a separate offline phase. Based on `CARL`, huge amounts of meta-data could be collected that will enable the

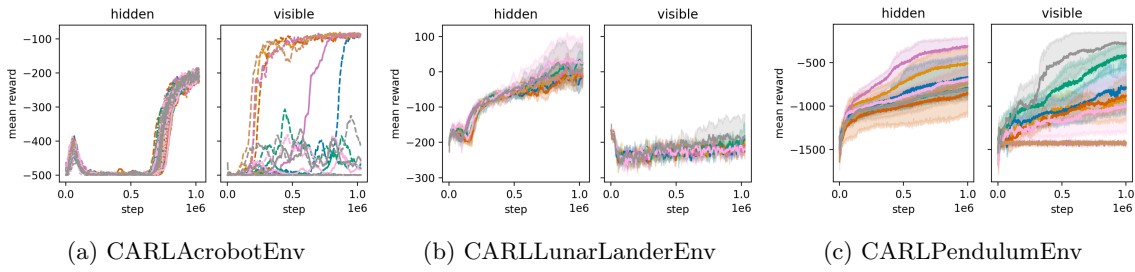

(a) CARLAcrobotEnv          (b) CARLLunarLanderEnv          (c) CARLPendulumEnv

Figure 24: Hyperparameter Optimization with PB2 (J. Parker-Holder et al., 2020). Hidden means that the context is hidden, and visible means that the full context is appended to the observation.

community to make use of classical meta-algorithmic approaches such as algorithm selection (Rice, 1976) for selecting previously learned policies or learning approaches.

### G.2    Challenge II: Uncertainty of RL Agents

With access to context information, we are able to study the influence of noise on RL agents in a novel way. While prior environments enabled studies on the behavior of agents when they could not be certain about their true state in a particular environment, the framework of Contextual Reinforcement Learning further allows studying agents' behaviors in scenarios with uncertainty on their current contextual environment, e.g., because of noise on the context features. In the practical deployment of RL, this is a reasonable concern since context features have to be measured somehow by potentially noisy sensors. As this setting affects the overall transition dynamics, Contextual Reinforcement Learning provides a unique test-bed in which the influence of uncertainty can be studied and how RL agents can deal with such. This line of research can also be combined with the work on unsupervised RL (Laskin et al., 2021; Schubert et al., 2023), where the agent learns a good policy initialization during a unsupervised pretraining phase, followed by a finetuning phase where the agent is optimizing an external reward. CARL enables researchers to either guide the pretraining process via uncertainty measures that are based on the context information or evaluate the robustness of finetuned policies under context noise.

### G.3    Challenge III: Interpretable and Explainable Deep RL

Trust in the policy is a crucial factor, for which interpretability or explainability often is mandatory. With the provided ground truth through the explicit use of context features, Contextual Reinforcement Learning could be the base for studying the interpretability and explainability of (deep) RL. By enabling AutoRL studies and different representation learning approaches, Contextual Reinforcement Learning will contribute to better interpreting the training procedures.

Contextual Reinforcement Learning further allows studying explainability on the level of learned policies. We propose to study the sensitivity of particular policies to different types of contexts. Thus, the value and variability of a context might serve as a proxy to explain the resulting learned behavior. Such insights might then be used to predict how policies might look or act (e.g., in terms of frequency of action usage) in novel environments, solely based on the provided context features.

### G.4    Challenge IV: AutoRL

AutoRL (Parker-Holder et al., 2022) addresses the optimization of the RL learning process. To this end, hyperparameters, architectures or both of agents are adapted either on the fly (Jaderberg et al., 2017a; Franke et al., 2021) or once at the beginning of a run (Runge et al., 2019). However, as AutoRL typically requires large compute resources for this procedure, optimization is most often done only on a per-environment basis. It is reasonable to assume that such hyperparameters might not transfer well to unseen environments, as the learning procedures were not optimized to be robust or to facilitate generalization, but only to improve the reward on a particular instance.

As we have shown above in Appendix F, Contextual Reinforcement Learning provides an even greater challenge for AutoRL methods. On the other hand, as CARL provides easy-to-use contextual extensions of a diverse set of RL problems, it could be used to drive research in this open challenge of AutoRL. First of all, it enables a large scale-study to understand how static and dynamic configuration approaches complement each other and when one approach is to be preferred over another. Such a study will most likely also lead to novel default hyperparameter configurations that are more robust and tailored to fast learning and good generalization. In addition, it will open up the possibility to study whether it is reasonable to use a single hyperparameter configuration or whether a mix of configurations for different instances is required (Xu et al., 2010). Furthermore, with the flexibility of defining a broad variety of instance distributions for a large set of provided context features, experiments with CARL would allow researchers to study which hyperparameters play a crucial role in learning general agents similar to studies done for supervised machine learning (van Rijn & Hutter, 2018) or AI algorithms (Biedenkapp et al., 2018).

### G.5 Challenge V: High Confidence Generalization

The availability of explicit context enables tackling another challenge in the field of safe RL. High Confidence Generalization Algorithms (HCGAs) (Kostas et al., 2021) provide safety guarantees for the generalization of agents in testing environments. Given a worst-case performance bound, the agent can be tasked to decide whether a policy is applicable in an out-of-distribution context or not. This setting is especially important for the deployment of RL algorithms in the real world where policy failures can be costly and the context of an environment is often prone to change. Contextual Reinforcement Learning has the potential to facilitate the development of HCGAs that base their confidence estimates on the context of an environment.

## H Context Features for Each Environment

We list all registered context features with their defaults, bounds and types for each environment family in Table 3 (classic control), Table 4 (box2d), Table 5 (brax) and Table 7 (RNA and Mario).

Table 3: Context Features: Defaults, Bounds and Types for OpenAI gym's Classic Control environments (Brockman et al., 2016)

(a) CARLCartPoleEnv

| Context Feature | Default | Bounds | Type |
|---|---|---|---|
| force_magnifier | 10.00 | (1, 100) | int |
| gravity | 9.80 | (0.1, inf) | float |
| initial_state_lower | -0.10 | (-inf, inf) | float |
| initial_state_upper | 0.10 | (-inf, inf) | float |
| masscart | 1.00 | (0.1, 10) | float |
| masspole | 0.10 | (0.01, 1) | float |
| pole_length | 0.50 | (0.05, 5) | float |
| update_interval | 0.02 | (0.002, 0.2) | float |

(b) CARLPendulumEnv

| Context Feature | Default | Bounds | Type |
|---|---|---|---|
| dt | 0.05 | (0, inf) | float |
| g | 10.00 | (0, inf) | float |
| initial_angle_max | 3.14 | (0, inf) | float |
| initial_velocity_max | 1.00 | (0, inf) | float |
| l | 1.00 | (1e-06, inf) | float |
| m | 1.00 | (1e-06, inf) | float |
| max_speed | 8.00 | (-inf, inf) | float |

(c) CARLMountainCarEnv

| Context Feature | Default | Bounds | Type |
|---|---|---|---|
| force | 0.00 | (-inf, inf) | float |
| goal_position | 0.50 | (-inf, inf) | float |
| goal_velocity | 0.00 | (-inf, inf) | float |
| gravity | 0.00 | (0, inf) | float |
| max_position | 0.60 | (-inf, inf) | float |
| max_position_start | -0.40 | (-inf, inf) | float |
| max_speed | 0.07 | (0, inf) | float |
| max_velocity_start | 0.00 | (-inf, inf) | float |
| min_position | -1.20 | (-inf, inf) | float |
| min_position_start | -0.60 | (-inf, inf) | float |
| min_velocity_start | 0.00 | (-inf, inf) | float |

(d) CARLAcrobotEnv

| Context Feature | Default | Bounds | Type |
|---|---|---|---|
| initial_angle_lower | -0.10 | (-inf, inf) | float |
| initial_angle_upper | 0.10 | (-inf, inf) | float |
| initial_velocity_lower | -0.10 | (-inf, inf) | float |
| initial_velocity_upper | 0.10 | (-inf, inf) | float |
| link_com_1 | 0.50 | (0, 1) | float |
| link_com_2 | 0.50 | (0, 1) | float |
| link_length_1 | 1.00 | (0.1, 10) | float |
| link_length_2 | 1.00 | (0.1, 10) | float |
| link_mass_1 | 1.00 | (0.1, 10) | float |
| link_mass_2 | 1.00 | (0.1, 10) | float |
| link_moi | 1.00 | (0.1, 10) | float |
| max_velocity_1 | 12.57 | (1.257, 125.7) | float |
| max_velocity_2 | 28.27 | (2.827, 282.7) | float |
| torque_noise_max | 0.00 | (-1.0, 1.0) | float |

Table 4: Context Features: Defaults, Bounds and Types for OpenAI gym's Box2d environments (Brockman et al., 2016)

(a) CARLBipedalWalkerEnv

| Context Feature | Default | Bounds | Type |
| --- | --- | --- | --- |
| FPS | 50.00 | (1, 500) | float |
| FRICTION | 2.50 | (0, 10) | float |
| GRAVITY_X | 0.00 | (-20, 20) | float |
| GRAVITY_Y | -10.00 | (-20, -0.01) | float |
| INITIAL_RANDOM | 5.00 | (0, 50) | float |
| LEG_DOWN | -0.27 | (-2, -0.25) | float |
| LEG_H | 1.13 | (0.25, 2) | float |
| LEG_W | 0.27 | (0.25, 0.5) | float |
| LIDAR_RANGE | 5.33 | (0.5, 20) | float |
| MOTORS_TORQUE | 80.00 | (0, 200) | float |
| SCALE | 30.00 | (1, 100) | float |
| SPEED_HIP | 4.00 | (1e-06, 15) | float |
| SPEED_KNEE | 6.00 | (1e-06, 15) | float |
| TERRAIN_GRASS | 10.00 | (5, 15) | int |
| TERRAIN_HEIGHT | 5.00 | (3, 10) | float |
| TERRAIN_LENGTH | 200.00 | (100, 500) | int |
| TERRAIN_STARTPAD | 20.00 | (10, 30) | int |
| TERRAIN_STEP | 0.47 | (0.25, 1) | float |
| VIEWPORT_H | 400.00 | (200, 800) | int |
| VIEWPORT_W | 600.00 | (400, 1000) | int |

(b) CARLLunarLanderEnv

| Context Feature | Default | Bounds | Type |
| --- | --- | --- | --- |
| FPS | 50.00 | (1, 500) | float |
| GRAVITY_X | 0.00 | (-20, 20) | float |
| GRAVITY_Y | -10.00 | (-20, -0.01) | float |
| INITIAL_RANDOM | 1000.00 | (0, 2000) | float |
| LEG_AWAY | 20.00 | (0, 50) | float |
| LEG_DOWN | 18.00 | (0, 50) | float |
| LEG_H | 8.00 | (1, 20) | float |
| LEG_SPRING_TORQUE | 40.00 | (0, 100) | float |
| LEG_W | 2.00 | (1, 10) | float |
| MAIN_ENGINE_POWER | 13.00 | (0, 50) | float |
| SCALE | 30.00 | (1, 100) | float |
| SIDE_ENGINE_AWAY | 12.00 | (1, 20) | float |
| SIDE_ENGINE_HEIGHT | 14.00 | (1, 20) | float |
| SIDE_ENGINE_POWER | 0.60 | (0, 50) | float |
| VIEWPORT_H | 400.00 | (200, 800) | int |
| VIEWPORT_W | 600.00 | (400, 1000) | int |

(c) CARLVehicleRacingEnv

| Context Feature | Default | Bounds | Type |
| --- | --- | --- | --- |
| VEHICLE | 0 | - | categorical |

Table 5: Context Features: Defaults, Bounds and Types for Google Brax environments (Freeman et al., 2021)

(a) CARLAnt

| Context Feature | Default | Bounds | Type |
|---|---|---|---|
| actuator_strength | 300.00 | (1, inf) | float |
| angular_damping | -0.05 | (-inf, inf) | float |
| friction | 0.60 | (-inf, inf) | float |
| gravity | -9.80 | (-inf, -0.1) | float |
| joint_angular_damping | 35.00 | (0, inf) | float |
| joint_stiffness | 5000.00 | (1, inf) | float |
| torso_mass | 10.00 | (0.1, inf) | float |

(b) CARLHalfcheetah

| Context Feature | Default | Bounds | Type |
|---|---|---|---|
| angular_damping | -0.05 | (-inf, inf) | float |
| friction | 0.60 | (-inf, inf) | float |
| gravity | -9.80 | (-inf, -0.1) | float |
| joint_angular_damping | 20.00 | (0, inf) | float |
| joint_stiffness | 15000.00 | (1, inf) | float |
| torso_mass | 9.46 | (0.1, inf) | float |

(c) CARLFetch

| Context Feature | Default | Bounds | Type |
|---|---|---|---|
| actuator_strength | 300.00 | (1, inf) | float |
| angular_damping | -0.05 | (-inf, inf) | float |
| friction | 0.60 | (-inf, inf) | float |
| gravity | -9.80 | (-inf, -0.1) | float |
| joint_angular_damping | 35.00 | (0, inf) | float |
| joint_stiffness | 5000.00 | (1, inf) | float |
| target_distance | 15.00 | (0.1, inf) | float |
| target_radius | 2.00 | (0.1, inf) | float |
| torso_mass | 1.00 | (0.1, inf) | float |

(d) CARLGrasp

| Context Feature | Default | Bounds | Type |
|---|---|---|---|
| actuator_strength | 300.00 | (1, inf) | float |
| angular_damping | -0.05 | (-inf, inf) | float |
| friction | 0.60 | (-inf, inf) | float |
| gravity | -9.80 | (-inf, -0.1) | float |
| joint_angular_damping | 50.00 | (0, inf) | float |
| joint_stiffness | 5000.00 | (1, inf) | float |
| target_distance | 10.00 | (0.1, inf) | float |
| target_height | 8.00 | (0.1, inf) | float |
| target_radius | 1.10 | (0.1, inf) | float |

(e) CARLHumanoid

| Context Feature | Default | Bounds | Type |
|---|---|---|---|
| angular_damping | -0.05 | (-inf, inf) | float |
| friction | 0.60 | (-inf, inf) | float |
| gravity | -9.80 | (-inf, -0.1) | float |
| joint_angular_damping | 20.00 | (0, inf) | float |
| torso_mass | 8.91 | (0.1, inf) | float |

(f) CARLUr5e

| Context Feature | Default | Bounds | Type |
|---|---|---|---|
| actuator_strength | 100.00 | (1, inf) | float |
| angular_damping | -0.05 | (-inf, inf) | float |
| friction | 0.60 | (-inf, inf) | float |
| gravity | -9.81 | (-inf, -0.1) | float |
| joint_angular_damping | 50.00 | (0, 360) | float |
| joint_stiffness | 40000.00 | (1, inf) | float |
| target_distance | 0.50 | (0.01, inf) | float |
| target_radius | 0.02 | (0.01, inf) | float |
| torso_mass | 1.00 | (0, inf) | float |

Table 6: Context Features: Defaults, Bounds and Types for Google Deepmind environments (Tassa et al., 2018)

(a) CARLDmcWalkerEnv

| Context Feature | Default | Bounds | Type |
|---|---|---|---|
| actuator_strength | 1.00 | (0, inf) | float |
| density | 0.00 | (0, inf) | float |
| friction_rolling | 1.00 | (0, inf) | float |
| friction_tangential | 1.00 | (0, inf) | float |
| friction_torsional | 1.00 | (0, inf) | float |
| geom_density | 1.00 | (0, inf) | float |
| gravity | -9.81 | (-inf, -0.1) | float |
| joint_damping | 1.00 | (0, inf) | float |
| joint_stiffness | 0.00 | (0, inf) | float |
| timestep | 0.00 | (0.001, 0.1) | float |
| viscosity | 0.00 | (0, inf) | float |
| wind_x | 0.00 | (-inf, inf) | float |
| wind_y | 0.00 | (-inf, inf) | float |
| wind_z | 0.00 | (-inf, inf) | float |

(b) CARLDmcQuadrupedEnv

| Context Feature | Default | Bounds | Type |
|---|---|---|---|
| actuator_strength | 1.00 | (0, inf) | float |
| density | 0.00 | (0, inf) | float |
| friction_rolling | 1.00 | (0, inf) | float |
| friction_tangential | 1.00 | (0, inf) | float |
| friction_torsional | 1.00 | (0, inf) | float |
| geom_density | 1.00 | (0, inf) | float |
| gravity | -9.81 | (-inf, -0.1) | float |
| joint_damping | 1.00 | (0, inf) | float |
| joint_stiffness | 0.00 | (0, inf) | float |
| timestep | 0.01 | (0.001, 0.1) | float |
| viscosity | 0.00 | (0, inf) | float |
| wind_x | 0.00 | (-inf, inf) | float |
| wind_y | 0.00 | (-inf, inf) | float |
| wind_z | 0.00 | (-inf, inf) | float |

(c) CARLDmcFingerEnv

| Context Feature | Default | Bounds | Type |
|---|---|---|---|
| actuator_strength | 1.00 | (0, inf) | float |
| density | 5000.00 | (0, inf) | float |
| friction_rolling | 1.00 | (0, inf) | float |
| friction_tangential | 1.00 | (0, inf) | float |
| friction_torsional | 1.00 | (0, inf) | float |
| geom_density | 1.00 | (0, inf) | float |
| gravity | -9.81 | (-inf, -0.1) | float |
| joint_damping | 1.00 | (0, inf) | float |
| joint_stiffness | 0.00 | (0, inf) | float |
| limb_length_0 | 0.17 | (0.01, 0.2) | float |
| limb_length_1 | 0.16 | (0.01, 0.2) | float |
| spinner_length | 0.18 | (0.01, 0.4) | float |
| spinner_radius | 0.04 | (0.01, 0.05) | float |
| timestep | 0.00 | (0.001, 0.1) | float |
| viscosity | 0.00 | (0, inf) | float |
| wind_x | 0.00 | (-inf, inf) | float |
| wind_y | 0.00 | (-inf, inf) | float |
| wind_z | 0.00 | (-inf, inf) | float |

(d) CARLDmcFishEnv

| Context Feature | Default | Bounds | Type |
|---|---|---|---|
| actuator_strength | 1.00 | (0, inf) | float |
| density | 5000.00 | (0, inf) | float |
| friction_rolling | 1.00 | (0, inf) | float |
| friction_tangential | 1.00 | (0, inf) | float |
| friction_torsional | 1.00 | (0, inf) | float |
| geom_density | 1.00 | (0, inf) | float |
| gravity | -9.81 | (-inf, -0.1) | float |
| joint_damping | 1.00 | (0, inf) | float |
| joint_stiffness | 0.00 | (0, inf) | float |
| timestep | 0.00 | (0.001, 0.1) | float |
| viscosity | 0.00 | (0, inf) | float |
| wind_x | 0.00 | (-inf, inf) | float |
| wind_y | 0.00 | (-inf, inf) | float |
| wind_z | 0.00 | (-inf, inf) | float |

Table 7: Context Features: Defaults, Bounds and Types for RNA Design (Runge et al., 2019) and Mario Environment (Awiszus et al., 2020; Schubert et al., 2021)

(a) CARLRnaDesignEnv

| Context Feature | Default | Bounds | Type |
|---|---|---|---|
| mutation_threshold | 5 | (0.1, inf) | float |
| reward_exponent | 1 | (0.1, inf) | float |
| state_radius | 5 | (1, inf) | float |
| dataset | eterna | - | categorical, $n = 3$ |
| target_structure_ids | f(dataset) | (0, inf) | list of int |

(b) CARLMarioEnv

| Context Feature | Default | Bounds | Type |
|---|---|---|---|
| level_index | 0 | - | categorical, $n = 15$ |
| noise | f(level_index, width, height) | (-1, 1) | float |
| mario_state | 0 | - | categorical, $n = 3$ |
| mario_inertia | 0.89 | (0.5, 1.5) | float |

