# OpenReview forum: "Contextualize Me – The Case for Context in Reinforcement Learning"
_TMLR — Accepted by TMLR_

### Review · Reviewer_jbAE · 2023-03-28

**Summary Of Contributions:**

This paper formalizes Contextual RL (cRL) in order to study agents' abilities to generalize across MDPs where certain properties of the MDP vary. An example from the empirical investigation is an agent's ability to generalize across instances of the Cartpole domain where the length of the pole is randomly assigned at the beginning of each episode. Agents are then provided "context" features which define the components of the MDP that are parameterized---in the above example, the agent is provided a "pole-length" feature. The paper provides theoretical and empirical support for the need for contextual information on a certain class of MDPs (called cMDPs).

**Audience:**

Yes

**Claims And Evidence:**

Yes

**Requested Changes:**

Changes I view as necessary:

1. The stated contributions need to clarify what this paper uniquely contributes beyond the current state of the field.
1. A more in-depth discussion about the relationship between cMDPs and POMDPs is needed, especially towards the goal of dynamic contexts and life-long learning.
1. Reconsider the formulation of cMDPs, perhaps instead writing the context as belonging to the state.
1. Rework the proof of proposition 1, clarifying what it means for states across different MDPs to communicate. Or, preferably, using the context-as-state formulation instead.
1. Change Figure 5 to show regret (or similar). Find a near-optimal policy for each configuration, then show the difference in performance between the SAC agent and the near-optimal performance.
1. Early in the paper, the concept of optimality gap is defined. Utilizing this measure in the experiments section would resolve several issues of confounding variables. Perhaps optimality gap should be reported for most (if not all) figures. Figures 5 and 8 appear to have the greatest need (i.e. the claims cannot currently be supported).
1. If only using 10 seeds, use a better aggregate statistic than IQM. It isn't clear that throwing away the top 2-3 and bottom 2-3 performing agents is justified in this setting---there is no reason to believe these are noisy samples or corrupt. I believe this ultimately hides information. Personally, I prefer the mean since it matches the evaluation proposed on page 5 eqn 1. At the very least, Agarwal et al. 2021 should be cited as I suspect that's where the paper got this choice. Also, use more than 10 seeds; the distributions in Figure 7 are _wild_.

Changes I view as improving the work:

1. Clean up some of the excess ink in all figures. For example, remove the grey background lines from Figure 5. Label all axes instead of a subset, or alternatively stack the plots in a neater way to share axes.
1. The robot example on page 2 does provide some clarity on terms. However, the example relies on context variables that are likely impossible to measure (such as friction). As such, this rather seems an example of a general POMDP and not an example of a cMDP.
1. On page 3 it is claimed "Context enables agents to learn more general policies". The opposite appears to be true as stated in the conclusion. Besides, providing additional context information can allow learning more discriminatory policies (using the context information to select which policy is used) suggesting exactly the opposite as well. Better ability to discriminate, not better ability to generalize.
1. Page 4: the frame-stacking example does not seem to fit the context definition. It is not clear how providing a history relates to providing a context. Instead frame-stacking appears to provide access to second order information, such as velocity.
1. Page 5: to estimate the optimality gap, the paper suggests learning an agent per context in order to approximate the best achievable performance. This either limits the formalization of cRL to discrete contexts, or adds a source of variance to the optimality gap metric. This limitation should likely be discussed.

**Strengths And Weaknesses:**

The benchmark provided by this paper can provide a nice testbed to understand how to best utilize "context" information. The empirical results suggest that simply concatenating this context information to the feature vector is insufficient. The paper highlights that there is clear future work needed to improve the use of these special features and provides a good tool for approaching that future work.

There are several issues throughout the paper, however, that detract from its utility in its current state. A majority of the listed contributions are not necessarily unique to this work. The formalization of cRL seems to inhibit many of the claims made throughout the paper. And the empirical evidence does not provide sufficient support for claims.

## Contributions
In the contributions list, the paper states:
> In short, our contributions are: (i) We provide a theoretical and empirical characterization of Contextual Reinforcement Learning

However, clearly Hallak et al., 2015 and Modi et al., 2018 both provide theoretical characterizations of cRL (the sole contribution of the Hallak paper is this theoretical characterization). The formalization in this paper does not appear to differ from Hallak's. I'm not sure this can be claimed as a contribution.

Additionally:
> [...] (iii) We demonstrate that context-oblivious agents are not suitable to solve contextual environments on their own.

Likewise, this was shown as the primary contribution of Malik et al. 2021, so again I doubt this should be claimed as a contribution of this work. Perhaps a better qualified version of this claim can be supported, since this paper does provide novel empirical evidence.

This work is also extremely similar to the work of Whiteson, Tanner, Taylor and Stone, 2011. In that work, they similarly discuss issues of generalization across environments where certain components of the environment are varied. Some examples include varying gravity in Mountain Car or joint length in Acrobot. In that work, the context information is discussed as forming a POMDP because the context was not provided to the agent. The work of Whiteson et al. should definitely be cited in this work.

In total, of the three stated contributions (i) and (iii) are entirely subsumed by prior works and (ii) is mostly subsumed by Whiteson et al. The primary novelty, I believe, is in applying the methodology of Whiteson et al. to a larger body of domains and comparing the performance of agents that include context information as part of state.

## Formalization of cRL

I understand the motivation to separate the context from the state, yielding a set of MDPs $(S, A, T_c, R_c, \rho_c)$ indexed by context $c$. However, this is really no different than a single MDP $(S \times C, A, T, R, \rho)$ where the context is part of the state-space. In prior works (i.e. Hallak et al.) the context is known ahead of time and is static for all time. In this case, there can never be communication between states in context $c_1$ and states in context $c_2$. In that work, it makes sense to treat each as a (nearly) independent MDP.

The example of a cMDP given at the bottom of page 2 mentions that context can change over time. Later in the middle of page 3, it is mentioned that contexts can aid in life-long learning. These two claims are both strictly at odds with the given formalization of cMDPs. By treating the cMDP as a set of MDPs indexed by $c$, the context must be static and known at the beginning of time. In order to accommodate a context that is dynamic, the context must be included in the state-space yielding a single valid MDP.

The equivalent context-as-state formalization is fairly straightforward. The transition matrix $T$ is now $(|S| \times |C|) \times (|S| \times |C|)$ as opposed to being a tensor of size $|S| \times |S| \times |C|$. In the case that the context is static, then we know that $T$ has at least $|C|$ unique communication classes which is what allows modeling this as $|C|$ different MDPs, with each $T_c$ being a valid stochastic matrix. However in the case that context is dynamic, then $T$ may have fewer than $|C|$ communication classes which means the system can transition from one context to the next. When $T$ is a tensor of valid stochastic matrices, there can be no communication between contexts (otherwise the matrices are not self-stochastic).

Finally, by including context as state, it is clear that a cMDP with the context being hidden from the agent is really a specific form of POMDP (as noted in appendix A). In this case, the partially observable component is the context. From here, it is already well-established that a deterministic optimal policy may not exist in the presence of hidden information.

## Proposition 1
The proof here is a bit unclear. It is stated that $\pi^*_{c1}$ is defined only on MDP $M_{c1}$. Then the proof reasons about the optimality of $\pi^*_{c1}$ on $M_{c2}$. However, this appears a contradiction: $\pi^*_{c1}$ is not defined on $M_{c2}$.

This proof also reasons about states from one MDP that are reachable from another MDP. In the given formulation where the cMDP is a set of MDPs, this logic does not make sense. Instead, this proof appears to require the context-as-state formulation discussed above. As written, I argue this portion of the proof is vacuously true rendering the proposition not useful.

## Section 5.1
This experiment is intended to study the generalization properties of a SAC agent when certain attributes of the environment change. However, changing attributes of the environment also changes the potential maximum achievable return in that environment. From figure 5, it is not possible to determine if the observed results are due to changing the maximum achievable return or due to the agent failing to generalize.

Whiteson et al. have a nice discussion of a related issue. They use Mountain Car as an example. If the force of the car is changed to be too small, or the gravity too large, then it is possible to make Mountain Car no longer solveable --- the agent simply cannot achieve high enough velocity to overcome the hill. The benchmarks throughout this paper can likely suffer from this same limitation. Notably the first subfigure of Figure 5 (max_speed) suggests that the blue lines are unable to solve the domain. Is this because of the agent failing to generalize, or is this because the max_speed is now too low that no agent can solve the environment?

## Section 5.3
The `Concat-all` agent is adding several bias units to the learning process, as several of the context variables never change. It isn't too surprising, then, that this harms out-of-sample generalization as there are several collinear features.

I'm curious how confidence intervals were computed in Figure 7. Clearly the distributions are highly skewed and/or multimodal, which makes computing confidence intervals challenging. It seems worth pointing out that the performance between no-context and context is identical on Pendulum in Figure 7. This suggests the variable being modified in the benchmark has little effect.

## Section 5.4
A prominent conclusion of these experiments is that neither agent generalizes particularly well. It appears the claim:
> [...] the visible-context agent can sometimes generalize better [than the hidden-context agent] to out-of-distribution contexts.

is not supported by the evidence in Figure 8. In all three columns, it seems rather clear that the hidden-context agent outperforms the visible-context agent for a substantially larger surface area.

## Questions

1. What is the x-axis in Figure 7 (labeled $k$)? Likewise the y-axis (labeled $R$)?
1. In Figure 7, how are confidence intervals calculated and about what statistic (e.g. percentile bootstrap about the mean)?

---

> ### Author Response · Authors · 2023-04-06
> **Initial Response 1**
>
> Thank you for taking the time to provide us with such in-depth feedback, we appreciate it.
>
> ## Contributions & Prior Work
> Here we specifically demarcate our work compared to Whiteson et al. (2011), whose work we now discuss in the update Related Work section in the paper.
> In their work Whiteson et al. focus on the evaluation protocol of general reinforcement learning (RL). In order to avoid overfitting on particular training environments they argue for generalized methodologies where they assess the performance of an agent on a set or distribution of environments. cMDPs can be formulized as a generalized environment, however, Whiteson et al. do not define the notion of context but just speak of general distributions. We deem the latter point important to derive meaning of generalization performances: With the definition of context, and especially context features which have physical meaning and are human-interpretable, we remove the abstraction of generalization and are able to precisely assess where agents perform well and where they do not.
> In addition, in their experiments they do provide information how their environments vary to the oracle agents by setting the agent’s or algorithm’s hyperparameters accordingly. This is in contrast to adding context to the state or as a special input from which the agent needs to learn.
> Also, Whiteson et al. *do not* vary any parametrization of the environment, especially nothing related to physical parameters, and they *do not* provide a benchmark as a test bed as we do.
> Therefore, we believe that with CARL and our notion of context we add a valuable contribution to the community.
>
> ## Formalization of cRL
> > A more in-depth discussion about the relationship between cMDPs and POMDPs is needed, especially towards the goal of dynamic contexts and life-long learning.
>
> > Reconsider the formulation of cMDPs, perhaps instead writing the context as belonging to the state.
>
> The question of how to solve this extended cMDP setting is secondary to this benefit in our opinion. Of course, we can simply treat a cMDP like a standard MDP or a POMDP when solving it, appending all available context information to the state or not showing it at all. The state should include information that we estimate to be important for solving an environment, after all, this certainly applies to context information. Seeing as many state features will change on a different timescale than context features do (e.g. during a single episode) and are often also necessary to solve single instances of the context space. We separate the categories of context features and state features to highlight the different semantic meaning and additional utility context features have, e.g. in curriculum learning or policy selection as shown by related work.
>
>
> ## Proposition 1
> We do not reason about reaching one state from one MDP from another in another MDP, that might have been a miscommunication, we updated the wording. Instead, we argue about acting in one MDP reaching a state also included in another MDP.
> As $M_{c1}$ and $M_{c2}$ together form the cMDP, they can share the same states. Let $s \in \mathcal{S}$ be a state from $M_{c1}$. We describe the three possible cases for state intersections of $M_{c1}$ and $M_{c2}$: 1. We can reach $s$ in $M_{c2}$, then $\pi^*_{c_1}$ can be optimal on that as well (no conflict) or is not optimal (conflict, we need a different policy to solve $M_{c2}$). 2. We cannot reach $s$ in $M_{c2}$, there we also need a new policy or at least extend $\pi^*_{c_1}$ because $\pi^*_{c_1}$ is not defined on the states of $M_{c2}$. Extending $\pi^*_{c_1}$ in this case does not remove the optimality on $M_{c1}$.
>
> [Whiteson et al., 2011] Whiteson, Shimon, Brian Tanner, Matthew E. Taylor, and Peter Stone. 2011. “Protecting against Evaluation Overfitting in Empirical Reinforcement Learning.” In 2011 IEEE Symposium on Adaptive Dynamic Programming and Reinforcement Learning (ADPRL), 120–27.

---

> > ### Author Response · Authors · 2023-04-06
> > **Initial Response 2**
> >
> > ## Potential Maximum Return on Different Contexts
> > While it is true that the context can cap the maximum return for an instance, we can clearly see this is not the case for most context features in Figure 5. Since this plot shows the percentage of evaluations below or at a given return on the x-axis, if the line is below 100% up until the maximum return, some portion of the evaluations have in fact reached the maximum possible reward. An example of a context feature limiting the maximum reward can be seen for the pole length 0.1*l.
> >
> > >Change Figure 5 to show regret (or similar). Find a near-optimal policy for each configuration, then show the difference in performance between the SAC agent and the near-optimal performance.
> >
> > In light of the agent still solving all instances of most context features at least some of the time, we’d like to ask how important you think this change is given it would add a significant amount of computing expense. Since the optimal reward would not be different for the SAC agent and the vast majority of agents shown in Figure 5, we believe this would simply add the same information we show in Figure 6.
> >
> > ## Optimality Gap & Reporting
> >
> >
> > >Early in the paper, the concept of optimality gap is defined. Utilizing this measure in the experiments section would resolve several issues of confounding variables. Perhaps optimality gap should be reported for most (if not all) figures. Figures 5 and 8 appear to have the greatest need (i.e. the claims cannot currently be supported).
> >
> >
> > At this moment, unfortunately, we are only able to approximate the optimality gap by training a single agent on a single context and comparing those to the general agent. Doing this for every experiment would increase the number of training runs and thus the compute from 100-1000, depending on the number of contexts.
> >
> > >If only using 10 seeds, use a better aggregate statistic than IQM. It isn't clear that throwing away the top 2-3 and bottom 2-3 performing agents is justified in this setting---there is no reason to believe these are noisy samples or corrupt. I believe this ultimately hides information. Personally, I prefer the mean since it matches the evaluation proposed on page 5 eqn 1. At the very least, Agarwal et al. 2021 should be cited as I suspect that's where the paper got this choice. Also, use more than 10 seeds; the distributions in Figure 7 are wild.
> >
> > For section 5.2 (optimality gap) we use 10 seeds and 10 test episodes for each of the 64 contexts and therefore 10*10*64 for calculating the IQM. Likewise, for section 5.4 (generalization) we have 10 seeds * 5 eval episodes per context thus 50 samples.
> > The rliable framework by Agarwal et al., 2021 now is rightfully cited. We agree that showing a certain statistics paints a certain picture which is why we show all three, mean, IQM and median, for section 5.2.
> > We also agree that the distributions in Figure 7 do look wild. This is due to the fact that some agents fail to learn anything meaningful, as we now show in Appendix Figure 19. Adding more seeds only marginally shrinks the confidence interval because the same behavior holds. Reporting on the stability of algorithms, i.e. analyzing how often an agent fails to learn, might be an important measure for future works in general.
> >
> > ## Questions
> > > What is the x-axis in Figure 7 (labeled )? Likewise the y-axis (labeled)?
> >
> > Figure 7 (now Figure 8) is now correctly labeled: x-axis environment steps, y-axis return.
> >
> > > In Figure 7, how are confidence intervals calculated and about what statistic (e.g. percentile bootstrap about the mean)?
> >
> > The 95%-confidence intervals for the mean are calculated with 1000 bootstraps samples as implemented in the seaborn plotting library. We updated the caption to be more verbose.

---

> > > ### Author Response · Authors · 2023-04-06
> > > **Initial Response 3**
> > >
> > > ## Suggestions
> > > > Figure 5
> > >
> > > We updated Figure 5 (now Figure 6) to use a log scale x-axis.
> > >
> > > > Robot example
> > >
> > > We agree that friction most likely is difficult to measure directly. However this still serves as an example for a cMDP which can be also formalized as a POMDP. Another example could be the provided payload (weight at the end effector) as context which we can measure. We updated the example.
> > >
> > > > On page 3 it is claimed "Context enables agents to learn more general policies". The opposite appears to be true as stated in the conclusion. Besides, providing additional context information can allow learning more discriminatory policies (using the context information to select which policy is used) suggesting exactly the opposite as well. Better ability to discriminate, not better ability to generalize.
> > >
> > > We agree and updated our wording. By a more discriminative policy we also obtain a more general policy, a policy which is applicable to more contexts.
> > >
> > > > Frame Stacking
> > >
> > > Frame stacking partially encodes the dynamics of the environment and thus yields information about second order quantities like velocity as rightly mentioned. But because in a cMDP we can have different transition dynamics per context/MDP this can also implicitly encode the context or differentiate the MDP we are currently in.
> > >
> > > > Estimation of the Optimality Gap
> > >
> > > The approximation of the Optimality Gap by training specialized agents on contexts sampled from the context distribution does induce a source of variance. The more contexts we sample the better our approximation.  We updated our wording accordingly. However, this approximation does not limit cRL to discrete contexts.

---

> > > > ### Comment · Reviewer_jbAE · 2023-04-13
> > > > **Looks good!**
> > > >
> > > > I read the above changes and they look good.
> > > >
> > > > Very minor comment: It still appears that the OG approximation is limited to discrete contexts, since we need to train a baseline for each context. We can only have a discrete number of baselines, so then we can only have a discrete number of estimates of OG.

---

> > > ### Comment · Reviewer_jbAE · 2023-04-13
> > > **Optimality Gap**
> > >
> > > It isn't clear to me how these percentages are showing solvability? This is showing that an agent can perform as well as another agent, which still allows both agents to be failing! There is no grounding in this case.
> > >
> > > As an example, consider the LunarLander domain. I could easily define a gravity constant such that the maximum thrust of the lifters cannot overcome gravity. This is still a physically plausible system (i.e. no negative gravity or anything), and I can still have agents that receive a variety of rewards e.g. because it is more rewarding to crash between the flags than to crash outside the flags, but still all agents are crashing. I want to know if some combinations of physical constants lead to impossible-to-solve problems and I don't _think_ the current analysis allows us to know this. On top of that, comparisons are made across contexts where the measurement units (return) are likely not comparable.
> > >
> > > > In light of the agent still solving all instances of most context features at least some of the time
> > >
> > > I don't think this is established, and I currently doubt that it is true. The CDF plots cannot show whether this is true unless the optimal return is included as a baseline when computing the CDF. Otherwise, the CDF only reasons about the performance of one agent w.r.t. all other agents without grounding.
> > >
> > > > Since the optimal reward would not be different for the SAC agent and the vast majority of agents shown in Figure 5
> > >
> > > I also am not sure that this is established. Though perhaps more precise language is required. The optimal _return_ should likely not be equivalent. Changing the maximum velocity in pendulum should change the theoretical minimum number of steps to the goal, no? Are you using discounted returns? If so, then shouldn't the theoretical maximum return be different?
> > >
> > > ---
> > >
> > > I'm curious, why introduce optimality gap if even the proposed approximation to OG is still too expensive to measure? Given the lack of grounding of Figure 6 (i.e. an estimate of the true maximum achievable return), it seems that Fig 6 still suffers from major confounding factors.
> > >
> > > ---
> > >
> > > I should warn that these are not `10 * 10 * 64` **independent** random samples. In fact, there is like a very large degree of correlation due to using multiple evaluations of the same policy. Using aggregate statistics like mean, median, or IQM is not quite correct with dependent samples. One alternative strategy is to use a stratified resampling approach both when computing aggregate statistics and when computing percentile bootstrap CIs.
> > >
> > > I do appreciate the use of bootstrap CIs here. Besides the independence issue, this is the right choice.

---

> > > > ### Author Response · Authors · 2023-04-14
> > > > **Response: Optimality Gap**
> > > >
> > > > ## Optimality Gap
> > > > >It isn't clear to me how these percentages are showing solvability? This is showing that an agent can perform as well as another agent, which still allows both agents to be failing! There is no grounding in this case.
> > > >
> > > > >I don't think this is established, and I currently doubt that it is true. The CDF plots cannot show whether this is true unless the optimal return is included as a baseline when computing the CDF. Otherwise, the CDF only reasons about the performance of one agent w.r.t. all other agents without grounding.
> > > >
> > > > We agree that also here our language lacked precision and it is true that the term “solvability”, as we used it, referred to the default agent (trained on the default context) as the reference point, which can of course still be failing. In the example in our paper which we refer to, the default agent has not failed, however. Judging by public benchmarking results (e.g. the [Open RL benchmark](https://wandb.ai/openrlbenchmark/sb3) or from [stable baselines zoo](https://github.com/araffin/rl-baselines-zoo/blob/master/benchmark.md)), the agents perform close to the accepted standard of good performance on a single context.
> > > >
> > > > Unfortunately there is no direct way to calculate the maximum reward possible depending on the context. The eCDF plots serve as a footprint of the severity of changing context.
> > > > We updated our language to better reflect your point and that we cannot reason about reductions to the theoretically achievable reward in this way.
> > > >
> > > >
> > > > >As an example, consider the LunarLander domain. I could easily define a gravity constant such that the maximum thrust of the lifters cannot overcome gravity. This is still a physically plausible system (i.e. no negative gravity or anything), and I can still have agents that receive a variety of rewards e.g. because it is more rewarding to crash between the flags than to crash outside the flags, but still all agents are crashing. I want to know if some combinations of physical constants lead to impossible-to-solve problems and I don't think the current analysis allows us to know this. On top of that, comparisons are made across contexts where the measurement units (return) are likely not comparable.
> > > >
> > > > We agree that such an analysis would be highly interesting, but without a perfect solver of these environments, this is, in our opinion, hard to accomplish. Even if extreme physical conditions do not lead to completely impossible to solve instances, they would require very specialized policies. In this case, it will be hard to tell at which point an agent fails to learn because the context cannot physically be solved -- the problem might simply become very hard and thus lead to failure to learn.
> > > >
> > > > >I also am not sure that this is established. Though perhaps more precise language is required. The optimal return should likely not be equivalent. Changing the maximum velocity in pendulum should change the theoretical minimum number of steps to the goal, no? Are you using discounted returns? If so, then shouldn't the theoretical maximum return be different?
> > > >
> > > > >I'm curious, why introduce optimality gap if even the proposed approximation to OG is still too expensive to measure? Given the lack of grounding of Figure 6 (i.e. an estimate of the true maximum achievable return), it seems that Fig 6 still suffers from major confounding factors.
> > > >
> > > > There are settings where the optimality gap can be estimated by other means than training an agent, especially if only the reward function is modified by the context (e.g. maze-like environments where we can directly search for the shortest path). In these cases, estimating the OG is significantly less costly.
> > > >
> > > > While we still maintain that approximating the OG for every experiment would be too costly, we can include the OG for this specific experiment. Just to clarify, however: do you mean the original Figure 6 (comparing general and specialized agents on CartPole and Pendulum) or the current Figure 6 (evaluating a specialized agent on different Pendulum contexts)?
> > > >
> > > > >I should warn that these are not 10 * 10 * 64 independent random samples. In fact, there is like a very large degree of correlation due to using multiple evaluations of the same policy. Using aggregate statistics like mean, median, or IQM is not quite correct with dependent samples. One alternative strategy is to use a stratified resampling approach both when computing aggregate statistics and when computing percentile bootstrap CIs.
> > > >
> > > > We are currently working on using stratified resampling for our reporting. We estimate that we can update the paper to include it by the middle of next week. As soon as we have clarification on which experiment we should provide the OG for, we will also start working on this as well and hopefully provide the results in the same update.

---

> > > > ### Author Response · Authors · 2023-04-20
> > > > **Updated Plots**
> > > >
> > > > We have now updated the plots in figures 8+9 where we aggregate results across both contexts and runs to use stratified resampling (using the RLiable library). We see no qualitative difference in the results although absolute values and confidence intervals slightly differ from our original plotting method. We thank you again for pointing this out.

---

> > ### Comment · Reviewer_jbAE · 2023-04-13
> > **Prior work and Formalism**
> >
> > ### Whiteson et al.
> >
> > It's not quite true that Whiteson et al. do not vary physical constants, in their paper they give an example of hovering a helicopter and varying wind speed. In their experiments, they vary the amount of force that an action provides. Their entire purpose is using this variation over parameterized environments in order to evaluate generalization of agents, which appears also to be your goal. Unfortunately, their code repository is only available on the wayback-machine, the original website is now down and the only public code implementation of their benchmark that I can find is my own!
> >
> > Nonetheless, I do agree that the benchmark that you provide is novel in that nearly all of the environments that you modify have not been modified in this way previously. However, I maintain that the idea of varying physical constants is not new and the idea of using context is not new. This is not meant to diminish the contribution of this work (the synthesis of old ideas can itself be novel and quite impactful!), however, it is useful to be precise about novelty.
> >
> > I continue to believe that this statement from review is true:
> > > The primary novelty, I believe, is in applying the methodology of Whiteson et al. to a larger body of domains and comparing the performance of agents that include context information as part of state.
> >
> > ### Formalism
> >
> > Hmm, I think we miscommunicated. I also am uninterested in discussing how to solve cMDPs, but rather am concerned about the formalization of the problem setting. The rest of your comment, though, is centered on challenges in solution-space (such as whether to append features, or if features change at different timescales). I agree these solution-space challenges are irrelevant to your paper, so aren't worth further discussion. There is some imprecision in your comment, such as:
> > > The state should include information that we estimate to be important for solving an environment
> >
> > In the formal definition of an MDP, there is no concept of estimation of state information. Some of the language in the paper---as well as this comment---confuses states with features or observations.
> >
> > I want to highlight two statements from my original review which were centered on the problem setting:
> > * Because context is not part of state, it makes no sense to talk about lifelong learning or contexts which change over time. If the context changes, then the set-of-mdps formalism that you adopt would require that the agent transition from one MDP to another MDP; which is problematic. If the context is part of state, then a changing context is equivalent to changing state. The current problem formalism does not well model your desired problem settings.
> > * If we consider the more general context-as-state formalism, then the claim in Theorem 1 is already well-established. We already know that this problem setting is partially observable when part of the state is occluded (the context, here) and a deterministic optimal policy is not guaranteed.
> >
> > ### Prop 1
> >
> > Okay, I accept the premise here. This did not make sense to me previously because this proof reasons about some state S = {S_c1, S_c2, S_c3} which is an alias for many different states, each indexed by their context. In this case, the term "state" threw me off as I would call S_c1 a state and S an aliased state (or an observation, perhaps). Reasoning about reachability of S is sensible (as the proof does), reasoning about reachability of S_c1 from M_c2 is not sensible (as I thought the proof was doing).

---

> > > ### Author Response · Authors · 2023-04-14
> > > **Response: Prior Work (Whiteson et al.) & Formalism**
> > >
> > > Thank you for your response and further comments!
> > >
> > > ## Whiteson et al.
> > > >It's not quite true that Whiteson et al. do not vary physical constants, in their paper they give an example of hovering a helicopter and varying wind speed. In their experiments, they vary the amount of force that an action provides. Their entire purpose is using this variation over parameterized environments in order to evaluate generalization of agents, which appears also to be your goal. Unfortunately, their code repository is only available on the wayback-machine, the original website is now down and the only public code implementation of their benchmark that I can find is my own!
> > >
> > > We apologize, you are correct that Whiteson et al. (2011) mention the Generalized Helicopter Environment [Koppejan & Whiteson, 2009]. We were referring to their experiments which do not use the helicopter environment but environments with generalization over non-physical based variations. Nevertheless, we should have been more thorough in this point and we now cite Koppejan & Whiteson when describing our notion of context in Section 2 “What is Context”. Thanks for pointing this out.
> > >
> > > ## Formalism
> > > >In the formal definition of an MDP, there is no concept of estimation of state information. Some of the language in the paper---as well as this comment---confuses states with features or observations.
> > >
> > > Thank you for noting that. Of course, we agree that it is important to be precise in our language. Therefore we checked our usage of state and observation/features and changed our text and the statement you cited above accordingly.
> > >
> > > We would like to defer to your statements from your original review regarding the problem setting.
> > >
> > > > Because context is not part of state, it makes no sense to talk about lifelong learning or contexts which change over time. If the context changes, then the set-of-mdps formalism that you adopt would require that the agent transition from one MDP to another MDP; which is problematic. If the context is part of state, then a changing context is equivalent to changing state. The current problem formalism does not well model your desired problem settings.
> > >
> > > We agree that for lifelong learning and similar paradigms, transitioning between MDPs would be necessary in the cMDP framework. As a first step, we believe the most relevant use cases of cMDPs are in the realms of zero- and few-shot generalization where this is not a problem. We have updated the paper to remove the references we had to scenarios which would require transitioning between MDPs.
> > >
> > > > If we consider the more general context-as-state formalism, then the claim in Theorem 1 is already well-established. We already know that this problem setting is partially observable when part of the state is occluded (the context, here) and a deterministic optimal policy is not guaranteed.
> > >
> > > We agree with your point here. Through previous iterations of this work, we have come to understand that while such a conclusion is indeed established, it still might not be very much accepted in the generalization community due to varying connotations of context in the literature. Thus, by explicitly stating this, we emphasize this conclusion for the case of cMDPs and provide concrete evidence.

---

### Review · Reviewer_jRh1 · 2023-03-29

**Summary Of Contributions:**

This paper studies the problem of changing dynamics and rewards in RL. They cast the problem of generalization as a contextual MDP. Then they go on to create a new suite of benchmark tasks where the context can be varied via interpretable quantities. Finally, their experiments show that generalizing RL agents to new contexts is a challenging open problem.

**Audience:**

Yes

**Broader Impact Concerns:**

I do not have any ethical concerns about this paper.

**Claims And Evidence:**

No

**Requested Changes:**

1. Please describe how the contextual MDP formulation is different from Parameterized Skills framework.
2. Please comment on the significance of your theoretical contribution.
3. If a context-aware agent does not outperform a context-oblivious one, consider dropping the claim that adding context is needed to adjust to task variations.
I am happy to be told that I have misunderstood any of the 3 aforementioned points. However, based on my current understanding, all 3 changes are critical to securing my recommendation.

**Strengths And Weaknesses:**

Strengths:
- Generalization to task variations is important for continual learning in RL
- The proposed benchmark could be valuable to the community because it is easier and more interpretable than procedurally generated environments like ProcGen and NetHack.

Weaknesses:
- In motivating cMDPs, the comparison to HiP-MDPs, Goal MDPs and Block MDPs was useful. However, I am struggling to understand how the cMDP formulation is different from Parameterized Skills [1]---as far as I understand, Parameterized Skills also seek to solve a distribution of MDPs which vary in transition and reward functions; furthermore, they also try to maximize return in expectation over all task variation. Could you please clarify how the cMDP formulation is different?
- The first contribution of the paper is supposed to be theoretical. My understanding is that the theoretical contribution is Theorem 1 which roughly states that the optimal policy in a contextual MDP needs to take context as input. This seems trivially true? If you need context to describe the transition and reward functions of the MDP, then you definitionally need it to describe the optimal policy.
- The 3rd stated contribution: "We demonstrate that context-oblivious agents are not suitable to solve contextual environments on their own." But by the authors' own admission, the empirical results do not support this claim --- the agent that has access to the context does *not* outperform the agent that is context-oblivious.

Question:
Meta Learning algorithms like MAML have also shown that policies can be adapted to new task instances. Can you also discuss how the cMDP framework differs? Would it make sense to compare cMDP approaches to Meta Learning?

[1] Learning Parameterized Skills, Bruno Castro da Silva et al, ICML 2012

---

> ### Author Response · Authors · 2023-04-06
> **Initial Response**
>
> Thank you for your review, we are grateful for your feedback.
>
> ## Difference to Parameterized Skills
>
> >In motivating cMDPs, the comparison to HiP-MDPs, Goal MDPs and Block MDPs was useful. However, I am struggling to understand how the cMDP formulation is different from Parameterized Skills [1]---as far as I understand, Parameterized Skills also seek to solve a distribution of MDPs which vary in transition and reward functions; furthermore, they also try to maximize return in expectation over all task variation. Could you please clarify how the cMDP formulation is different?
>
> The Parameterized Skills paper [Da Silva et al., 2012] does not formalize the setting as an MDP, which is why we use the definition by Hallak et al. (2015), even though the setting they describe is quite similar (obviously apart from the task of mapping specialized policies to instances). Therefore we updated our related work accordingly including Da Silva et al. (2012) together with Hallak et al. (2015) for this formalization.
> The subtle differences are (i) the caveat that they ‘assume that the MDPs have dynamics and reward functions similar enough so that they can be considered variations of a same task’ and (ii) they define the learning goal as maximizing the expected return on a task times its sampling probability with no mention of a possible train or test setting. We believe that the cMDP framework is slightly more general and explicit in these aspects, thus we continue to credit Hallak et al. alongside Da Silva et al. here.
>
>
> ## Significance of the Theoretical Contribution
> >Please comment on the significance of your theoretical contribution.
> Our main motivation for including our theoretical findings is to illustrate the usefulness of context for generalization. Compared to more comprehensive results of subproblems of cMDPs, e.g. for policy selection in epistemic POMDPs [Ghosh et al., 2021], they may seem simple, but we do believe it is important to state them nonetheless. We hope the updated paper organization reflects their inclusion as a motivational tool.
> ## Empirical Results
> >The 3rd stated contribution: "We demonstrate that context-oblivious agents are not suitable to solve contextual environments on their own." But by the authors' own admission, the empirical results do not support this claim --- the agent that has access to the context does not outperform the agent that is context-oblivious.
> We strongly disagree with this characterization of both our claim and the reporting of our results. Our results do not provide strong support for a claim like “naive context-aware agents solve contextual environments”, but this is not what we are claiming at all. Rather we are showing that context-oblivious agents face a much harder learning problem and do not generally seem to be equipped to solve it (see Section TODO). We do not claim that context concatenation to the state is the missing ingredient - in fact our experiments suggest that it is not and this is what we are describing.
>
> [Da Silva et al., 2012] Da Silva, Bruno, George Konidaris, and Andrew Barto. 2012. “Learning Parameterized Skills.” arXiv [cs.LG]. arXiv. http://arxiv.org/abs/1206.6398.
>
> [Hallak et al., 2015] Hallak, Assaf, Dotan Di Castro, and Shie Mannor. 2015. “Contextual Markov Decision Processes.” arXiv [stat.ML]. arXiv. http://arxiv.org/abs/1502.02259.

---

> > ### Comment · Reviewer_jRh1 · 2023-04-13
> > **Follow up**
> >
> > I like that you have included more details about your benchmark environments in favor of your theoretical results. Also, thank you for correcting me about your third contribution: it would indeed be an interesting result if you could show that adding context in a naive way (concat) is *insufficient* for solving cMDPs.
> >
> > Questions:
> > 1. In Section 5.3, you hypothesize that the reason adding context is unhelpful in Pendulum is because of the increased size of the state-space. But, shouldn't this effect disappear in the long-run? In other words, if you run your experiment longer, wouldn't the agent learn to ignore the unhelpful parts of the input? If so, then the concat-all strategy might still give good asymptotic performance, which would be reassuring.
> > 2. Regarding da Silva's Parameterized Skills, you say that "The subtle differences are (i) the caveat that they ‘assume that the MDPs have dynamics and reward functions similar enough so that they can be considered variations of a same task’" Do cMDPs not make the same assumption? I understand your second point, they maximize expected reward over a task distribution and cMDPs decompose the problem into train vs test.
> > 3. Is it possible that you are not seeing the benefit of adding context in Pendulum because it is an easy problem? Do you observe similar results in other, more challenging domains?

---

> > > ### Author Response · Authors · 2023-04-14
> > > **Additional Questions**
> > >
> > > Thank you for your response and this active discussion.
> > >
> > > > 1.In Section 5.3, you hypothesize that the reason adding context is unhelpful in Pendulum is because of the increased size of the state-space. But, shouldn't this effect disappear in the long-run? In other words, if you run your experiment longer, wouldn't the agent learn to ignore the unhelpful parts of the input? If so, then the concat-all strategy might still give good asymptotic performance, which would be reassuring.
> > >
> > > We assumed this should be the case as well and ran this experiment for 10 times as many training steps (see performance over time [here](https://imgur.com/M2wl4rg), we plot mean and standard deviation). We still see no meaningful difference between the agents in this case. This could be partly due to the agent already performing quite well and the difficulty of the representation learning problem in this case.
> > > In results on HalfCheetah ([plot](https://imgur.com/3KocvbQ), mean and standard deviation), for example, we can see that the agent exposed to the whole context learns significantly slower to the degree where it cannot improve during training. We therefore believe this representation learning problem of correctly identifying relevant context information is very challenging and additional training time alone may not be enough to solve it.
> > >
> > > >2. Regarding da Silva's Parameterized Skills, you say that "The subtle differences are (i) the caveat that they ‘assume that the MDPs have dynamics and reward functions similar enough so that they can be considered variations of a same task’" Do cMDPs not make the same assumption? I understand your second point, they maximize expected reward over a task distribution and cMDPs decompose the problem into train vs test.
> > >
> > > cMDPs do not make an assumption of similarity between different contexts. This assumption is also rather arbitrary and vague, since defining similarity and what ‘a task’ is is hard. The only assumption for cMDPs is that we use the same action and observation space which we believe is a much more flexible approach.
> > >
> > > >3. Is it possible that you are not seeing the benefit of adding context in Pendulum because it is an easy problem? Do you observe similar results in other, more challenging domains?
> > >
> > > At this point, it is hard to reason about the relation of the simplicity of the environment, such as Pendulum, and the effect of concatenating context to the observation. For example, in the more complex walker environment, we see a benefit in concatenating the context. However, we also have results on HalfCheetah showing no difference between the hidden and concat agent (concatenating the changing context feature; concatenating all available ones prevents learning). We believe these diverging results are partly caused by the entanglement of the representation learning and generalization problem which poses an open challenge for the community.

---

> > > > ### Comment · Reviewer_jRh1 · 2023-05-01
> > > > **Additional Experiments**
> > > >
> > > > 1. HalfCheetah Experiment: thank you for including this. Here too it seems like adding context (even when you know the correct context variable, i.e, the case you call "non-static") does not significantly help in solving contextual MDPs.
> > > > 2. Do you have a sense for which contextual MDPs do benefit from additional context (even when the context variables are known)? I understand that not all environments benefit from additional context but perhaps you can share your insights about *when* it is helpful to include more context. Can you run non-static vs no-context agents on more problems in your proposed benchmark?

---

### Review · Reviewer_Psm1 · 2023-03-29

**Summary Of Contributions:**

In this paper, the authors study the problem of generalization in RL through the lens of contextual MDPs, that is, through contextual RL (cRL). In particular, the authors argue that this framework allows for a clear formalization of the problem from the point of view of the theory as well as from the point of view of the experimental setups. To this end, the authors propose theoretical arguments that show only a policy conditioned on context can be optimal in contextual MDPs. Additionally, the authors provide a suite of environments that come with a context, which usually is not the default in RL. The authors conduct experiments on some of these environment to try to validate the cRL approach.

**Audience:**

Yes

**Claims And Evidence:**

Yes

**Requested Changes:**

Could the authors spend more time on clearly presenting the benchmark?

Could the authors explain and situate the importance of the theoretical results?

Could the authors address the experimental results concerns previously mentioned?

**Strengths And Weaknesses:**

**Strengths**
- The problem of generalization is important in RL and we still don't fully grasp what it means, therefore the authors tackle an important problem.
- The authors do a job at presenting the problematic in the introduction and motivate the rest of the paper
- The authors study the effect of context in some of the environments in the benchmark that they propose

**Weaknesses**
- The important of the theoretical results are not clear and not clearly situated within the literature
- The organization of the paper could be improved. A lot of space is spent on theory, while the presentation of the environments is totally deferred to the appendix.
- The empirical results do not seem to corroborate the claims of the paper

The problem of generalization in RL is an important one, and also a fundamental one. In a sense, it is so fundamental that someone's definition of generalization may be quite different from another researcher. It all depends on the setting we care about. That being, the authors argue that cRL is a way to formalize this. I am not convinced that this is truly general. Indeed, cRL relies on a context. If this does not exist, where does the context come from? The authors make arguments in the end of section 2 that it could come from representation learning. However, then the problem is essentially one of learning representations and the benefit of the context is less clear.

To this end, the authors propose a benchmark for cRL. However, this takes about 20 lines in the main paper and all the details are relegated to the appendix. This is not great if the authors hope for the community to latch on such a benchmark. The presentation of all the environments should be clear and in the main paper in my opinion. The sections within the appendix refer to each other and we can get lost easily. In B.1, it is mentioned that "joint stiﬀness, gravity, friction, (joint) angular damping, actuator strength, torso mass as well as target radius and distance deﬁne the context." What is left as features that are not part of the context? It would be great if this was clearly stated.

Although space is an issue and it is hard to put everything in the main paper, I am not sure what the role of the theoretical section is in the paper. In particular, the main theoretical result seems not very surprising and I would be really interested in the authors clearly situating this result within the CMDP literature.

The empirical investigation is an important part of the paper as it illustrates the importance of the benchmark. However, looking closely at the results, it seems like there is not a very significant difference with the baselines. For example, in Figure 6bd, the x axis are "zoomed in" to make a point, but a difference of 20 points on Cartpole or even less on Pendulum is not that much. In the figure 7, it seems like there is little statistical difference between the methods, yet the text reads as "agents without access to context information are not able to solve even simple contextual environments" It seems like such claims could be rewritten to better reflect the results.

---

> ### Author Response · Authors · 2023-04-06
> **Initial Response**
>
> Thank you for your comments, we appreciate your feedback.
>
> ## Benefits of Context
>
> >Indeed, cRL relies on a context. If this does not exist, where does the context come from? The authors make arguments in the end of section 2 that it could come from representation learning. However, then the problem is essentially one of learning representations and the benefit of the context is less clear.
>
> Independent of how we provide the context features to the agent, context in a cMDP is first and foremost a tool for task specification. Context allows us to define training and test settings for RL agents precisely and with some degree of interpretability to humans. This in turn also lets humans interpret the performance of RL agents in terms of the context space (i.e. how quickly does performance degrade when the context space becomes larger or smaller? Which areas of the context space are difficult for the agent? Where does the agent perform well in the test setting?).
> Thus even if the current true context is unknown to the agent we can still compare performances across different context distributions. How we represent the context, i.e. whether we concatenate it to the state if it is known or we learn it from the state if it is unknown or not directly observable by the agent, is a subproblem of cRL.
>
> ## Paper Structure
> It is true that the theoretical results claim quite a lot of space in the main paper. We wanted to thoroughly motivate the use of context, but upon further consideration, we removed parts of this section in favor of extending the benchmark description in Section 4.
>
> ## Importance of Theoretical Results
> As stated above, our main motivation for including these theoretical findings is to illustrate the usefulness of context for generalization. Compared to more comprehensive results of subproblems of cMDPs, e.g. for policy selection in epistemic POMDPs [Ghosh et al., 2021], they may seem simple, but we do believe it is important to state them nonetheless. We hope the updated paper organization reflects the inclusion of our theoretical results as a motivational tool.
>
> ## Empirical Results
> >However, looking closely at the results, it seems like there is not a very significant difference with the baselines. For example, in Figure 6bd, the x axis are "zoomed in" to make a point, but a difference of 20 points on Cartpole or even less on Pendulum is not that much.
>
> While it is true that for Cartpole and Pendulum the return does not drastically deteriorate, the return distributions are in fact different and worse for the general agent. How much the return deteriorates depends on the environment.
>
> >In the figure 7, it seems like there is little statistical difference between the methods, yet the text reads as "agents without access to context information are not able to solve even simple contextual environments" It seems like such claims could be rewritten to better reflect the results.
>
> In the case of Figure 7, we used the term ‘solve’ because there is a solution threshold for these environments that the specialized agents meet and the general agent does not. However we agree that the wording evokes a strong and general image not capturing the nuances depending on the environment: On some environments the effect is way stronger and some it is not as pronounced. We updated the formulation.

---

> > ### Comment · Reviewer_Psm1 · 2023-04-14
> >
> > I would like to thank the authors for taking the time to improve their paper and write the rebuttal.
> >
> > I think that giving more space to the empirical results is a great improvement, as such a benchmark would be highly valuable for future work. The current work does present some arguments for why context is beneficial, but to me it seems like it still hasn't been convincingly demonstrated. The authors mention a few times that the benefits of using context will depend on environments, which inherently makes sense, but it would have been great to have a few examples as to how such benefits may happen and why. At the current state, this is not very clear to me as we don't have evidence to make a strong case for context, even if a limited domain of application was considered. In this sense, the benchmark is an important contribution so that future researchers investigate this question more deeply.

---

> > > ### Comment · Reviewer_Psm1 · 2023-05-01
> > >
> > > I would like to thank the authors for engaging actively in the discussions. The authors have significantly improved the paper from the initial version by adding more seeds and changing some experiments. However, an important issue to me is the illustrated by the experiments on CartPole. Initially there was no major difference between using context or not, whereas the second version of the experiments do show a difference. Despite this improvement, this highlights to me that context is not straightforward to benefit from. As such, proposing a benchmark of environments with context without a in depth empirical analysis leaves the question out as to whether the proposed context should help experiments. Usually in the literature, when a benchmark is proposed (e.g. Unsupervised RL Benchmark, D4RL), a set of baselines is validated on that benchmark. I would invite the authors to present at least one baseline on the set of environments that is proposed in order to evaluate more in-depth the importance of this benchmark.

---

### Author Response · Authors · 2023-04-06
**Central Response**

We thank all reviewers for their thoughtful and valuable feedback. We address individual comments in direct replies to the reviews, but want to give an overview of updates to the paper as well as some common points of the reviews.

## Updates to the Paper
As a result of the feedback received we made several updates to the paper:
additional clarifications with regards to related work as well as the addition of Whiteson et al. 2011
removing parts of the theory Section 2 and 3 in favor of extending the benchmark Section 4 by moving parts of the appendix
updated empirical results in Section 5.4
We mark update text with green color.

## Overlap with Previous Work
While it is true that previous work has used variations of the cMDP framework and provided both theoretical and empirical results, we believe our focus is different. We are interested in using the full cMDP framework, not a more constrained subproblem of it [Hallak et al., 2015; Ghosh et al., 2021], to measure an agent's generalization abilities. In contrast, much of the literature is concerned with solving cMDPs by making use of specialist policies instead [Da Silva et al., 2012; Hallak et al., 2015; Doshi-Velez & Konidaris, 2016; Perez et al., 2020; Ghosh et al., 2021], an approach we think is orthogonal and complimentary to the goal of more general RL agents.

We comment on specific works in the individual comments. To improve clarity and situate our work better in the literature as rightly commented, we improved our discussion of previous work on cMDPs.

## Contradictions between Theoretical and Empirical Findings
We agree that our empirical results, in particular Figure 8 in Section 5.4, unintuitively contradict our earlier statement that context is necessary to generalization. We attributed these results to the increased state space due to concatenating the context features combined with the large number of training contexts. One thing we missed was the interaction effect of the environment. On CartPole, gravity and the pole length are strongly linked: lower gravity combined with higher pole length will have the same effect on the policy as higher gravity combined with lower pole length because of the differential equations describing the underlying physics. In this case, context might not be necessary to learn a good policy but increases the state space that the agent’s function has to be fitted on.The same experiment with the update interval instead of the pole length reveals the magnitude of this effect. Now we can see better generalization from the agent receiving the context information in distribution A and C compared to hidden as well as different generalization patterns of both in B.
Both varying update interval and gravity paints a similar picture with a less pronounced effect.
Our results underline that providing context by concatenating it to the state *can* help generalization, but it is not guaranteed. It strongly depends on type of training distribution and the interaction of context features and demands further investigation.
Please find our updated discussion of these results in Section 5.4.

## References
[Da Silva et al., 2012] Da Silva, Bruno, George Konidaris, and Andrew Barto. 2012. “Learning Parameterized Skills.” arXiv [cs.LG]. arXiv. http://arxiv.org/abs/1206.6398.

[Doshi-Velez & Konidaris, 2016]  Doshi-Velez, F., & Konidaris, G. D. (2016). “Hidden parameter markov decision processes: A semiparametric regression approach for discovering latent task parametrizations.”, arXiv [cs.LG]. arXiv. https://arxiv.org/abs/1308.3513.

[Perez et al., 2020] Perez, C., Such, F., & Karaletsos, T. (2020). “Generalized Hidden Parameter MDPs:Transferable Model-Based RL in a Handful of Trials..” arXiv [cs.LG]. arXiv https://arxiv.org/abs/2002.03072.

[Hallak et al., 2015] Hallak, Assaf, Dotan Di Castro, and Shie Mannor. 2015. “Contextual Markov Decision Processes.” arXiv [stat.ML]. arXiv. http://arxiv.org/abs/1502.02259.

[Ghosh et al., 2021] Ghosh, Dibya, Jad Rahme, Aviral Kumar, Amy Zhang, Ryan P. Adams, and Sergey Levine. 2021. “Why Generalization in RL Is Difficult: Epistemic POMDPs and Implicit Partial Observability.” arXiv [cs.LG]. arXiv. http://arxiv.org/abs/2107.06277.

---

### Author Response · Authors · 2023-05-04
**Central Response: Baselines**

Thank you for your continued feedback on the paper. The latest round of comments has encouraged us to extend our experiments to the rest of the benchmark. We agree that providing a baseline for each environment is important and thus will start running a hidden agent and a concat-all agent for all environments. We believe these are important baselines for the development of better solution methods for cMDPs as they represent a context-agnostic and naively context aware method, both of which should be outperformed by any method tailored to solving cMDPs.
Additionally we aim to provide some estimation of the optimal performance via a set of specialized agents (as noted by Reviewer JbAE), though only for a selection of environments due to the large computational demands. As this is time and resource consuming, we will update the paper in batches, starting with the full set of classic control experiments for which we expect results around next week.

---

### Author Response · Authors · 2023-05-17
**Baseline Results & Paper Updates**

Dear Action Editor, dear Reviewers,

Thank you for your suggestions and comments. We have further improved the text, especially the abstract, introduction and conclusion to fit the previous changes. Additionally, we include benchmarking results on more environments, showing the benefits of context-aware agents on the majority of them. We hope to add the continuous control environments (DM control and Brax) as well as BipedalWalker within two weeks.

List of changes:
- Changes to the contributions of the paper, especially in abstract & introduction
- Move related work after section 2 (Contextual Markov Decision Process) and highlight work of Whiteson et al.
- Added baselines, the paper now includes all classic control environments, Lunar Lander and Mario
- Incorporated new results into Section 6.3

The most recent changes are marked blue in the text.

---

> ### Comment · Reviewer_jbAE · 2023-05-24
> **Updating review based on latest changes**
>
> With the modifications to the paper, I believe this paper is a good fit for TMLR as written. The concerns w.r.t. misaligned stated contributions have been resolved---the paper now provides clear contributions and evidence to support those contributions. The experiments have better statistical evidence supporting their claims and _mostly_ provide direct support. The benchmark itself is clearly novel and the paper states the intended audience and how the benchmark can be used to address an open problem that is of interest to at least some readers of TMLR.
>
> I believe there is still one flaw with the experiments that we have talked past each other on. I do not believe this flaw is enough to prevent publication, with the modified claims in the paper I think there is more than sufficient evidence regardless. However, for a maximally clear and polished paper, I think it is still worth reconsidering the eCDF plots in Figure 6. There is a fundamental issue when using empirical CDFs as a metric of performance: if all agents perform poorly, then there is still one agent that performs best. That is, we are considering only a ranking of agents without considering whether _any_ agent is actually performing well. There is no grounding.
>
> I recognize that computing or providing such grounding can be computationally expensive, maybe even impossible in some scenarios. However, the paragraph at the bottom of page 9 ignores that the units being compared are not truly comparable. By metaphor, the return that an agent receives in super-gravity Lunar Lander is not comparable to the return achieved in normal-gravity Lunar Lander. The paper currently presents this as a limitation of current solution strategies---certainly I agree that current solution strategies have room for improvement. But I maintain that some of the difference in observed performance is from changes in fundamental problem hardness. Differences in irreducible error, so to speak.
>
> **Edit:** The more I consider this issue, I realize that I would actually be quite content with a sentence or two in the paper addressing this as a limitation of Figure 6. Let the audience decide how important that limitation is while they consider your results. Concretely saying something like:
> > Note that by multiplying physical constants by $A$ likely changes the fundamental difficulty of each environment instance. Limiting the maximum speed, for instance, likely requires even an optimal agent to take more steps in order to achieve the goal state.
>
> -----
> The following are very pedantic notes while re-reading the current draft, they slightly impacted the clarity for me, but do not influence whether the paper should be published in TMLR.
>
> Notes:
> * The new version of the abstract looks really good now. The sentence: "We confirm the insight that optimal behavior in cRL requires context information, as in related areas like partial observability" doesn't quite make sense. PO doesn't necessarily require context information, rather it formalizes a problem setting where there is missing information. It is slightly more correct to say that cRL is a subset of PO with slightly more structure---we know what that missing information is and that it is statically determined at the beginning of time.
>
> * In the intro, it's not quite clear that RL has shown little success in real-world deployments that require generalization. Mostly, I'm unsure what this sentence means and I'm not sure there's evidence to support it. Perhaps a relevant citation or two to complement the citations that RL mostly has been used in narrow domains (though again, I'm not totally sure what those words mean either).

---

> > ### Author Response · Authors · 2023-05-26
> > **Thank you!**
> >
> > Thank you for your update and your acceptance recommendation. We agree that adding a note on the limitations of the experiment in Figure 6 is a good solution and will do so for the final revision. We'll also aim to incorporate your additional notes.
> > Beyond that, we would also like to thank your for your extensive feedback that significantly improved the paper throughout the review process.

---

### Decision · Action_Editors · 2023-05-25

**Recommendation:** Accept with minor revision

**Comment:**

Both the authors and reviewers should be commended for the effort put in during the discussion period. All three reviewers agreed on acceptance, and the paper clearly meets TMLRs criteria for publication.

The authors should consider reviewer jbAE's comment here (https://openreview.net/forum?id=Y42xVBQusn&noteId=wb6g5CSlg8)---in the end its a request for an additional line of text in the paper---and make any final edits.

**Audience:**

This paper studies the role of context in changing dynamics and rewards in reinforcement learning promoting. The papers advocates for using contextual MDP and introduces a new benchmark. The benchmark is novel and is a nice testbed to investigate how to best make use of context information. The experiments included are well done from a methodological point of view and illustrate both the challenges and potential benefits of utilising context information. There is a clearly of interest to the TMLR community.

**Claims And Evidence:**

The authors have made significant improvements both sharpening the claims in the paper and providing clear empirical evidence to support those claims including several new experiments during the author response period. All three reviewers agreed.

---

> ### Author Response · Authors · 2023-05-26
> **Thank you & Final Revisions**
>
> We would like to thank both the action editor and the reviewers for their engagement in the review process, helping us improve the paper and leading to this positive decision. We will incorporate the changes suggested by reviewer jbAE. We expect to upload out final revisions until Wednesday next week.